computational biology/biomathematics/
computer modelling and simulation

contact network, diffusion process, epidemiology, influenza, mathematical model

**Author for correspondence:**
Md Shahzamal
e-mail: md.shahzamal@students.mq.edu.au

# Indirect interactions influence contact network structure and diffusion dynamics

Md Shahzamal[1,2], Raja Jurdak[1,2], Bernard Mans[1] and Frank de Hoog[2]

[1]Department of Computing, Macquarie University, Sydney, Australia
[2]Data61, Commonwealth Scientific and Industrial Research Organization (CSIRO), Brisbane, Australia

MS, 0000-0003-4903-9531

Interaction patterns at the individual level influence the behaviour of diffusion over contact networks. Most of the current diffusion models only consider direct interactions, capable of transferring infectious items among individuals, to build transmission networks of diffusion. However, delayed indirect interactions, where a susceptible individual interacts with infectious items after the infected individual has left the interaction space, can also cause transmission events. We define a diffusion model called the same place different time transmission (SPDT)-based diffusion that considers transmission links for these indirect interactions. Our SPDT model changes the network dynamics where the connectivity among individuals varies with the decay rates of link infectivity. We investigate SPDT diffusion behaviours by simulating airborne disease spreading on data-driven contact networks. The SPDT model significantly increases diffusion dynamics with a high rate of disease transmission. By making the underlying connectivity denser and stronger due to the inclusion of indirect transmissions, SPDT models are more realistic than same place same time transmission (SPST)-based models for the study of various airborne disease outbreaks. Importantly, we also find that the diffusion dynamics including indirect links are not reproducible by the current SPST models based on direct links, even if both SPDT and SPST networks assume the same underlying connectivity. This is because the transmission dynamics of indirect links are different from those of direct links. These outcomes highlight the importance of the indirect links for predicting outbreaks of airborne diseases.

## 1. Introduction

Modelling diffusion processes on contact networks is an important research area for epidemiology, marketing, and cybersecurity.

In the diffusion processes, contagious items initially appear at one or more nodes of interacting systems and then spread over the system through inter-node transmissions occurring due to interactions among nodes. Thus, a wide range of research has been conducted to understand the co-relations between diffusion dynamics and underlying contact network properties which are defined by interaction patterns [1–4]. Most diffusion models assume both infected and susceptible individuals are simultaneously present in the same physical space (e.g. visiting a location) or virtual space (e.g. friendship in online social networks) for an inter-node transmission, called individual-level transmission, to occur. We denote these models as same place same time transmission-based diffusion (SPST diffusion) which only accounts for individual-level transmission links created by direct interactions to build underlying contact networks [5,6]. Examples of the SPST diffusion are message dissemination in Mobile Ad-hoc Networks [7–9], information diffusion in online social networks [10] and infectious disease spreading through physical contacts [11].

The focus on concurrent presence (real or virtual), however, is not sufficiently representative of a class of diffusion scenarios where inter-node transmissions can occur via indirect interactions, i.e. where susceptible individuals receive contagious items even if the infected individuals have left the interaction location. For example, an individual infected by airborne disease can release infectious particles in the air through coughing or sneezing. The particles are then suspended in the air so that a susceptible individual arriving after the departure of the infected individual can still get infected [5,12,13]. Similarly, a piece of information posted by an existing member in an online social blog can be seen by a newly joined member, even though the new member was not present when the piece of information was posted [14,15]. Queen message dissemination in the social ant colonies and pollen dissemination in the ecology also follow a similar mechanism [16]. In these scenarios, current diffusion models can miss significant transmission events during delayed indirect interactions.

We develop a diffusion model to capture individual-level transmission links created for both the direct interactions where susceptible and infected individuals are present at visited locations and indirect interactions where susceptible individuals are present or arrive after the infected individuals have left the locations. We call this model same place different time transmission-based diffusion (SPDT diffusion). In this model, links are created through location and time and called SPDT links. The SPDT model captures transmission events occurring simultaneously at multiple locations by an infected individual (transmission at the current location due to the direct interaction, and transmissions at the previous locations due to indirect interactions). In this diffusion model, the link infectivity, which is the probability of causing infection by a SPDT link which is created for direct and/or indirect interactions, depends on various factors such as the presence of infected and susceptible individuals at the interaction location, decay rates of infectious items and environmental conditions etc [5,12,17,18]. The current SPST models cannot account for these features as the inclusion of indirect interactions can significantly affect a diffusion process.

The SPDT diffusion model is integrated with a new assessment method for finding SPDT link infectivity. For an airborne disease, this involves two steps: (1) determining the number of infectious particles inhaled by the susceptible individual (intake dose) and (2) analytically finding the corresponding infection risk, probability of contracting disease. To assess the infection transmission probability of an interaction between susceptible and infected individuals in airborne disease, the Wells–Riley model and its variants are widely used [12,17,19]. These models can determine infection probability for a susceptible individual who is exposed to infectious particles generated by a number of infected individuals for a period of time at a visited location. They take into account the particle generation rates of infected individuals, the breathing rate of the susceptible individual and particle removal rates from the interaction location. However, these models do not support link infectivity assessment for the SPDT model as they do not consider arrival and departure times of individuals at interaction locations. Thus, they cannot account for transmission during indirect interactions. Our proposed SPDT link infectivity assessment method is based on the Wells–Riley model, yet it accounts for transmissions during indirect interactions among susceptible and infected individuals as well as the impacts of environmental and structural factors of interaction locations through particles' decay rates.

The diffusion behaviours of our proposed SPDT model are explored through simulating airborne disease spreading on empirical dynamic contact networks constructed from location updates of a social networking application called Momo [20]. In these networks, the disease transmission links are created for both direct and indirect co-located interactions. We analyse 56 million location updates from 0.6 million Momo users of Beijing city to extract all possible direct and indirect links. This yields a SPDT contact network (SDT) of 364K users and exclusion of indirect links from the above process provides a SPST contact network (SST) of the same users. These networks are based on the sparse

data as users were not regular in using the Momo app. Thus, the impact of indirect interactions can be stronger with denser data which is explored by building denser networks (DDT and DST) from the SDT network. For the inclusion of indirect interactions, the SPDT model adds new transmission links and new users to the SPDT networks over the SPST networks. To understand the enhancement in diffusion behaviours for these additions, other SPDT and SPST networks (LDT and LST) are built with the same users and same link densities of DDT network. We adopt a generic Susceptible–Infected–Recovered (SIR) epidemic model to simulate airborne disease spreading on these networks [21].

The SPDT model alters the underlying contact network dynamics (e.g. contact frequency and contact duration for adding new indirect links) which also vary with the infectivity decay rates of SPDT links. This variation in the underlying connectivity affects the diffusion dynamics [22,23]. The infectivity decay rates of SPDT links are strongly influenced by the infectious particle decay rates at the interaction locations. Therefore, we first investigate how various particle decay rates change the underlying connectivity of the SPDT model and hence diffusion dynamics, using known airborne disease parameters such as infectiousness of particles and infectious period of a disease [4,24–27]. We also study how the impacts of particle decay rates vary with biological disease parameters. Then, we compare the SPDT diffusion dynamics with SPST diffusion dynamics to identify novel behaviours that the SPDT model introduces. Finally, we show that the SPDT diffusion dynamics cannot be reproduced by current SPST models, even when controlling for link densities, and this characterizes the limitations of the underlying network connectivity in the SPST model compared to that of the SPDT model.

## 2. SPDT diffusion model

The infectious items transmission network in the SPDT model is built at the individual level with direct and (delayed) indirect transmission links created for direct and indirect interactions. The link creation procedure for this model can be explained by airborne disease spreading phenomena, as shown in figure 1. In this particular scenario, an infected individual A (host individual), red circle, arrives at location L, blue dashed circle, at time $t_1$ followed by the arrival of susceptible individuals u and v, green circles, at time $t_2$. The appearance of v at L creates a directed link for transmitting infectious particles from A to v and lasts until time $t_4$ making direct link during $[t_2, t_3]$ and indirect link during $[t_3, t_4]$. The indirect link is created as the impact of A still persists (as the virtual presence of A is shown by the dashed circle surrounding A) after it left L at time $t_3$, due to the survival of the airborne infectious particles in the air of L. But, the appearance of u has only created direct links from A to u during $[t_2, t_3]$. Another susceptible individual w arrives at location L at time $t_5$ and a link is created from A to w through the indirect interaction due to A's infectious particles still being active at L. However, the time difference between $t_5$ (arrival time of w) and $t_4$ (departure time of A) should be the maximum $\delta$ such that a significant particle concentration is still present at L after A left at $t_4$.

The infected individual makes several such visits, termed as active visits, to different locations and transmits disease to susceptible individuals through his infectious particles. The unique feature of the SPDT model compared with SPST models is that the infected individual can transmit disease at multiple locations in parallel due to direct transmission links at current locations and indirect transmission links at the previous locations that do not require the presence of the host individual. However, visits of infected individuals at locations where no susceptible individuals are present or susceptible individuals visit after the time period $\delta$ do not lead to disease transmissions. During the active visits, directed transmission links between infected individuals and susceptible individuals are created through location and time. We call these links SPDT links which can have components: direct transmission links and/or indirect transmission links. The disease transmission probability over a SPDT link is influenced by the indirect link duration along with direct link duration, the time delay between neighbour and host individual appearances at the interaction location and decay rates of infectious particles from the location [5,12,17].

We combine a method with the SPDT model to assess the probability of contracting disease through a SPDT link (also called SPDT link infectivity) using generic assumptions. This method captures the parallel disease transmission events of infected individuals. Suppose that an infected individual A appears at a location L at time $t_s$ and deposits airborne infectious particles into the air of L with a rate $g$ (particles/s) until he has left the location at $t_l$. These particles are homogeneously distributed into the air volume $V$ of proximity and the particle concentration keeps increasing until it reaches a steady

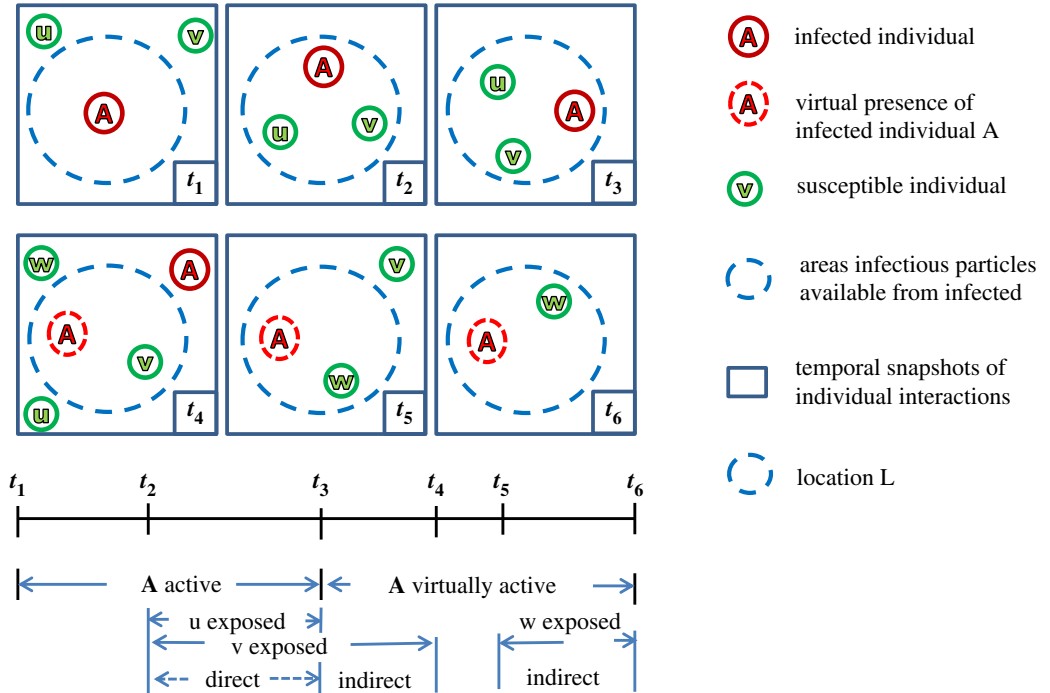

**Figure 1.** Disease transmission links creation for co-located interactions among individuals in SPDT model. The upper part shows the six snapshots of interactions over time at a location and the lower part shows the periods of exposure through direct and indirect interactions. Susceptible individuals are linked with the infected individual if they enter the blue dashed circle areas within which particles are available to cause infection.

state. Simultaneously, active particles decay at a rate $r$ (proportion/s) from the proximity due to various reduction processes such as air conditioning, settling down of particles to the ground and losing infectivity.

Thus, the particle concentration $C_t$ at time $t$ after the infected individual A arrives at L is given in [17,28]

$$C_t = \frac{g}{rV}(1 - e^{-r(t-t_s)}). \tag{2.1}$$

If a susceptible individual u (as figure 1) arrives at location L at time $t_s' \geq t_s$ and continues staying with A up to $t_l' < t_l$, the number of particles $E_d$ inhaled by u through this direct link is

$$E_d = \frac{gp}{rV}\left[t_l' - t_s' + \frac{1}{r}e^{-r(t_l'-t_s)} - \frac{1}{r}e^{-r(t_s'-t_s)}\right], \tag{2.2}$$

where $p$ is the pulmonary rate of u. If a susceptible individual v (as figure 1) stays with A as well as after A leaves L at $t_l$ where $(t_l' \geq t_l)$, it will have both direct and indirect links. The value of $E_d$ for v within the time $t_s'$ and $t_l$ is given by

$$E_d = \frac{gp}{rV}\left[t_l - t_s' + \frac{1}{r}e^{-r(t_l-t_s)} - \frac{1}{r}e^{-r(t_s'-t_s)}\right]. \tag{2.3}$$

For the indirect link from time $t_l$ to $t_l'$, we need to compute the particle concentration during this period which decreases after A leaves at $t_l$. The particle concentration at time $t$ is given by

$$C_t = \frac{g}{rV}(1 - e^{-r(t_l-t_s)})e^{-r(t-t_l)} \tag{2.4}$$

The individual v inhales particles $E_i$ during the indirect period from $t_l$ to $t_l'$ is

$$E_i = \frac{gp}{Vr^2}(1 - e^{-r(t_l-t_s)})[1 - e^{-r(t_l'-t_l)}] \tag{2.5}$$

If a susceptible individual w is only present for the indirect period at the proximity (as figure 1), the number of inhaled particles for the indirect period from $t'_s$ to $t'_l$ is given by

$$E_i = \frac{gp}{Vr^2}(1 - e^{-r(t_l - t_s)})[e^{-r(t'_s - t_l)} - e^{-r(t'_l - t_l)}] \tag{2.6}$$

Therefore, the total inhaled particles for an SPDT link are

$$E_l = \frac{gp}{Vr^2}[r(t_i - t'_s) + e^{rt_l}(e^{-rt_i} - e^{-rt'_l})] + \frac{gp}{Vr^2}[e^{rt_s}(e^{-rt'_l} - e^{-rt'_s})], \tag{2.7}$$

where $t_i$ is given as follows: $t_i = t'_l$ when the SPDT link has only direct component, $t_i = t_l$ if the SPDT link has both direct and indirect components, and otherwise $t_i = t'_s$. If $t_s < t'_s$, we have to set $t_s = t'_s$ for appropriate exposure. If a susceptible individual receives $m$ SPDT links from infected individuals during an observation period, the total exposure $E$ is

$$E = \sum_{k=0}^{m} E_l^k, \tag{2.8}$$

where $E_l^k$ is the received exposure for kth SPDT link which have direct and/or indirect components. The probability of the infection spreading can be determined by the dose–response relationship defined as

$$P_I = 1 - e^{-\sigma E}, \tag{2.9}$$

where $\sigma$ is the infectiousness of virus to cause infection [5,29,30]. This value depends on both the disease type and the infectiousness of particles.

# 3. Data and methods

## 3.1. Dataset

This study exploits location update information collected from users of a social discovery network Momo[1] [31]. The Momo app updates the current user locations to the Momo server while the app is used. The authors of [20] collected location updates from the server every 15 min over 71 days (from May to October 2012). Each database entry includes coordinates of the location, time of update and user ID. The app updates a user's location whenever the user moves at least 10 m. For this study, the updates from Beijing are used as it is the city with the highest number of updates for the period of 32 days from 17 September 2012 to 19 October 2012. These data contain almost 56 million location updates from 0.6 million users.

## 3.2. Contact networks

A data-driven individual dynamic contact network is built analysing location updates of Momo users collected from Beijing city over 32 days. This network includes possible direct and indirect transmission links due to direct and indirect co-location interactions among users. To create a SPDT link between a host user (assumed infected with an airborne disease) and a neighbouring user (assumed susceptible—not infected yet), the neighbouring user should make location updates within 20 m distance of the host user's current locations and updates must be made within 200 min of the last update of the host user from the current location. When the last update of the host user is made from more than 20 m distance of his first update of the current location or after 30 min of his immediate previous update, a new link creation process is started at the new location. However, the link creation is continued at the previous location for up to 200 min since the last update of the host user. If a host user does not have any neighbouring user over the observation period, he/she is not included in the constructed network. Therefore, the processing of 56 million location updates from 0.6 million users yields an SPDT contact network of 364 K users with a total of 6.86 million links. To compare SPDT diffusion to SPST diffusion, we generate a corresponding SPST network excluding the indirect links from the SPDT network.

The above constructed networks show low link densities as users often appear in the system for an average of 3–4 days and then disappear for the remainder of the data collection period. This is

[1]https://www.immomo.com.

characterized by the limitations of the collection system and user's behaviours when using the social networking app. Thus, these networks are called Sparse SPDT network and Sparse SPST network which capture partial snapshots of real-world social contact networks. In this paper, Sparse SPDT network is denoted as SDT network and Sparse SPST network as SST network. However, SPDT network means any network with both direct and indirect links, while SPST network mean network with direct links only. Therefore, these sparse networks might underestimate the diffusion dynamics because infected individuals may not be present in the network for all of their infectious periods and miss some disease transmission events. This may also lead to an incorrect conclusion regarding the contribution of SPDT diffusion model. To understand the SPDT diffusion in networks with high link densities, we reconstruct a Dense SPDT network (DDT network) repeating links from available days of a user to the missing days for that user [2,3]. In this process, all links of a day picked randomly from available days are copied to a random missing day. Thus, the DDT network has links for every day for each user. Then, the corresponding Dense SPST network (DST network) is built excluding indirect links from the DDT network.

The users who are only connected with other users through indirect links in the above SPDT networks become isolated in SPST networks as indirect links do not exist in SPST networks. Thus, link densities reduce in SPST networks compared to the corresponding SPDT networks. Moreover, the underlying social contact structures are also reshaped since some users disconnect from each other. This is the inherent property of SPDT model over SPST model and its impacts on diffusion dynamics are characterized as follows. We create two networks, LDT and LST, which maintain the same link densities and the same underlying social structure as that of the DDT network. In this format, neighbouring user's arrival time $t'_s$ of the SPDT links that have only indirect components in DDT network is set to the $t_s$ of the host user to obtain an LDT network. Then, indirect components of links are removed from the LDT network to build the LST network which now has the same link density and no isolated users. These two networks are used to identify the novel diffusion behaviours of the SPDT model compared to the SPST model while varying link densities and underlying social structures. All SPDT networks assume the same network structure (same clustering coefficient and degree distribution) than the SDT network, similarly all SPST networks structures are the same as the SST network.

## 3.3. Epidemic model

A generic SIR epidemic model is adapted to emulate airborne disease propagation on the constructed contact networks [21]. Individuals are in one of the three states: susceptible, infectious and recovered. If an individual in the susceptible state receives SPDT links from infected individuals, they may move to the infectious state with the probability derived by equation (2.9). Then, the infectious individual continues to produce infectious particles over its infectious period $\tau$ days until they enter the recovered state. In this epidemic model, no event of births, deaths or entry of new individual are considered.

## 3.4. Simulation set-up

The simulations are step forwarded in our experiments with a one-day interval. The authors of [2,3] have studied that aggregating contact information in one day provides similar disease spreading dynamics of considering each contact separately. Moreover, newly infected individuals in an influenza-like disease have an incubation period before becoming infectious [4]. Thus, the 1-day interval can be considered as a latent period. All simulations are run for a period of 32 days. Simulations from a single seed node, initially infected, requires a long time to produce a full epidemic curve (disease prevalence reaches to a peak value and then declines). Thus, we chose an initial set of 500 seed nodes randomly in each experiment to start simulations assuming that it will be capable of showing the full epidemic curve in 32 days. In addition, it is sufficient to demonstrate the contribution of indirect links within this setting. During each day of disease simulation, the received SPDT links for each susceptible individual from infected individuals are separated and infection causing probabilities are calculated by equation (2.9). The volume $V$ of proximity in equation (2.7) is fixed to $2512 \, \mathrm{m}^3$ assuming that the distance, within which a susceptible individual can inhale the infectious particles from an infected individual, is 20 m and the particles will be available up to the height of 2 m [5,32]. The other parameters are assigned as follows: particle generation rate $g = 0.304$ PFU (plaque-forming unit)/s and pulmonary rate $q = 7.5 \, \mathrm{l \, min}^{-1}$ [26,27,32]. Based on the air exchange rates 0.5–6 h$^{-1}$ in public settings

[33,34], infectivity decay rate of generated particles and settling rates [5], we assume particle removal rates are in the range $0.2$–$8\,h^{-1}$. Therefore, particles may require 7.5 min to 300 min to be removed from interaction areas after their generation. We assign $r = 1/60\,b$ to equation (2.7) where $b$ is particle removal time randomly chosen from [7.5–300] min given a median particle removal time $r_t$. The particle removal rates $r$ in our experiments are discussed with $r_t$, i.e. particle decay rates $r_t$ mean the corresponding particle removal rates $r$ drawn from the above process for all SPDT links and $r_t$ is median of $1/60r$. The parameter $\sigma$ is set to 0.33 as the median value of required exposures for influenza to induce disease in 50% susceptible individuals is 2.1 PFU [25]. Susceptible individuals stochastically switch to the infected states in the next day of simulation according to the Bernoulli process with the infection probability $P_I$ (equation 2.9). Individual stays infected up to $\tau$ days randomly picked up from 3 to 5 days maintaining $\bar{\tau} = 3$ days (except when other ranges are mentioned explicitly) [4].

## 3.5. Characterizing metrics

We have collected the following values of disease incidents at each day of simulation to characterize diffusion dynamics in the networks: number of infections $I_n(t)$ caused at a simulation day $t$, number of infected individuals $I_r(t)$ recovered from infection and current number of infected individuals $I_p(t)$ (disease prevalence) in the system on a simulation day $t$. For characterizing the diffusion dynamics, we find the disease reproduction abilities of infected individuals in a network as follows. The disease prevalence dynamics at time $t$ in the compartmental model is given by

$$\frac{\mathrm{d}I_p(t)}{\mathrm{d}t} = (\beta S(t) - \sigma)I_p(t), \tag{3.1}$$

where $\beta$ is the infection rate, $\sigma$ is the recovery rate, $S(t)$ is the number of susceptible individuals at $t$ [35]. If $\beta S(t) > \sigma$, the disease prevalence $I_p(t)$ gets stronger at time $t$. Otherwise, $I_p(t)$ reduces and disease dies out if it is continuously reduced. The ratio $\beta S(t)/\sigma$ is called effective reproduction number, the number of secondary infections caused by an infected individual based on the condition that all individuals in the network are not susceptible to the disease, and termed as $R_t$. Therefore,

$$R_t = \frac{\beta S(t)}{\sigma} = \frac{\beta S(t)I_p(t)}{\sigma I_p(t)} = \frac{I_n(t)}{I_r(t)}.$$

We can replace $\beta S(t)I_p(t)$ with the number of infections $I_n$ caused on a simulation day and $\sigma I_p(t)$ with the number of recovered individual from infection $I_r(t)$. Thus, we can estimate the effective reproduction number $R_t$ at each simulation day using $I_n$ and $I_r$. Then, we find the average $R_e$ of $R_t$ summing $R_t$ and dividing by the number of simulation days. The value of $R_e$ represents the overall strength of a disease to diffuse on the contact networks.

# 4. Results and analysis

The inclusion of indirect transmission links makes the underlying transmission network of the SPDT model strongly connected compared to the SPST model. Individuals who are not connected in the SPST model may get connected in the SPDT model for adding indirect transmission links. Hence, the underlying connectivity becomes denser in the SPDT model. Secondly, the number of links between two connected individuals may increase with the inclusion of indirect links. In addition, a direct link connecting two individuals in SPST models can be extended by appending an indirect transmission link in the SPDT model. These enhancements increase the disease transmission probabilities among individuals in the SPDT model. Therefore, the diffusion dynamics will be amplified in the SPDT model compared to the SPST model [23,36].

The SPDT model alters the network connectivity when changing the particle decay rates $r_t$ which influences the diffusion dynamics. To understand this, we consider figure 2 where the particle concentration rises and falls for various $r_t$ according to equation (2.1) and equation (2.4) at a location visited by an infected individual. Here, the particle concentration decays are estimated assuming that the infected individual has stayed in the visited location for 200 min. As the value of $r_t$ is increased, significant particle concentration is available for longer time at the interaction location (figure 2b). This allows more neighbouring individuals to receive exposure when visiting locations within the longer period $\delta$ at higher $r_t$, when indirect transmission links can be created. Therefore, the underlying

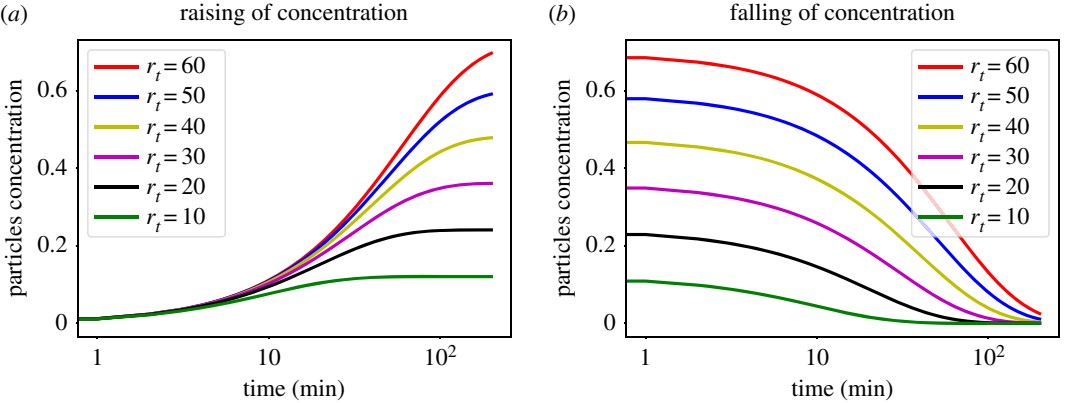

**Figure 2.** Particle concentrations for various particles decay rates $r_t$, where $r_t = 1/60r$ min and $r$ is the proportion of particles decayed per second, at a location when (*a*) an infected individual is present over 200 min and (*b*) the infected individual has left the location after staying 200 min

network connectivity becomes denser with increasing $r_t$. In addition, the exposure through direct and indirect links becomes stronger as $r_t$ increases (figure 2*a,b*). Thus, individuals in the SPDT model get connected more strongly at high $r_t$ and link infectivity increases. We, therefore, study the variations in the underlying network connectivity with $r_t$ and their impacts on diffusion.

The disease parameters are known to influence the spreading dynamics as well [24]. In our SPDT model definition, we find an interaction between the infectiousness of diseases and the particle decay rates (equation (2.9)). For example, the higher value of infectiousness $\sigma$ will increase the infection risk of SPDT links. Thus, the required threshold of the particle concentration to cause infection through a SPDT link reduces and the indirect link creation window $\delta$ will be longer. This means that more susceptible individuals will create links with the infected individual for visiting the same location at a higher $\sigma$. Therefore, the underlying connectivity gets stronger with high $\sigma$ for a given particle decay rate $r_t$. In addition, the other disease parameter, the infectious period $\tau$, also affects disease spreading varying the recovery rates of infected individuals. Thus, we study how the SPDT model behaves with stronger disease parameters and various $r_t$. Finally, we explore the novel behaviours SPDT model introduces through controlling link densities.

## 4.1. Network analysis

We analyse the enhancements in the underlying connectivity of the SPDT model exploring two network metrics: degree centrality and local clustering coefficient. We study the metrics for static and dynamic representations of sparse networks (SST and SDT networks) over 32 days. In static networks, an edge between two individuals is created once they have a link over 32 days. A SPDT link will be an edge when the inhaled exposure $E_l$ by the susceptible individual is $E_l \geq 0.01$ particles (as equation (2.9) shows $P_I$ negligible at $E_l = 0.01$). However, the edges are not weighted by $E_l$ or the number of SPDT links. We set $r$ in equation (2.7) corresponding to $r_t = 60$ min to understand the maximum enhancements by the SPDT model. We find the degree distribution, the number of neighbours a host individual has contacted, and the local clustering coefficient, which is the ratio of the number of triangles present among the neighbours and the possible maximum triangles among neighbours. To compute the clustering coefficient, we neglect the directions of links as our focus is to understand the changes in connectivity. The results are obtained using NetworkX [37] and are shown in figure 3*a,b* for the static network. In the SPDT network, the number of individuals with low degree decreases while the number of individuals with high degree increases compared to those of the SPST network (figure 3*a*). The same changes are found for the clustering coefficient as well (figure 3*b*).

The dynamic representations are created by aggregating networks over each day where an edge between two individuals is created once they have a link on that particular day. Then, we measure the daily average degree and clustering coefficient for the SPDT network with the values of $r_t = \{10, 20, 30, 40, 50, 60\}$ min and also for the SPST network. The results are shown in figure 3*c,d*. The daily average individual degree and average clustering coefficient in the SPDT model are significantly larger than that in the SPST model and the difference increases as $r_t$ increases. However, the increases in daily average degree and daily average clustering coefficient decrease at the higher

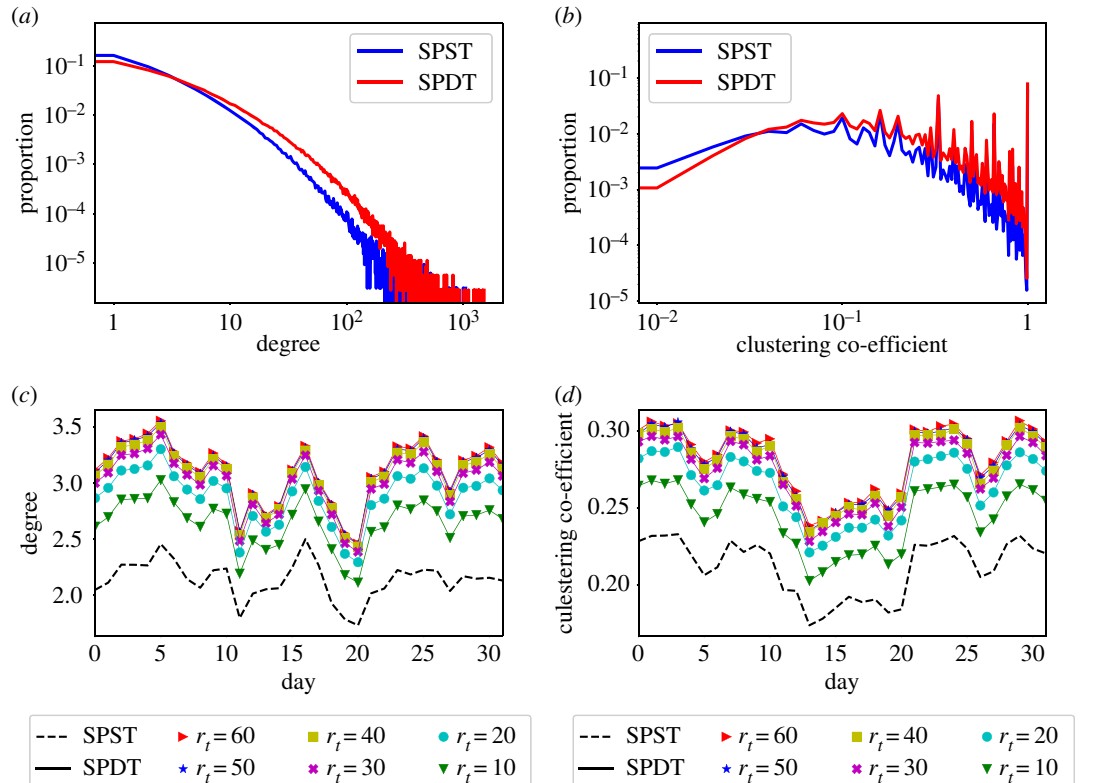

**Figure 3.** SPST (dashed line) and SPDT (solid lines) networks properties: (*a*) degree distribution in static networks, (*b*) clustering co-efficient distribution in static networks, (*c*) daily average degree in dynamic networks, and (*d*) daily average clustering co-efficient in dynamic networks.

values of $r_t$ as the particle concentration reaches a steady state quickly at high $r_t$. Both the static and dynamic networks show the stronger connectivity in the SPDT model than the SPST model and the network properties vary with particle decay rates $r_t$.

## 4.2. Diffusion for various particle decay rates

This experiment explores the influence of particle decay rates on diffusion dynamics varying $r_t$ in the range [10, 60] min with a step of 5 min. We run 1000 simulations for each $r_t$ on both SPDT and SPST networks with sparse and dense configurations. Figure 4 shows the overall diffusion characteristics for all $r_t$ where y-axes show the impacts on diffusion dynamics for corresponding $r_t$. In total, outbreak sizes (total infections caused over the simulation period) in the SPDT model increase linearly with $r_t$ (figure 4*a*). The amplification in outbreak size with the SPDT model is up to 5.6 times for sparse networks and 4.3 times for dense networks at $r_t = 60$ min (figure 4*b*). The individuals in the SPDT model achieve strong disease reproduction abilities $R_e$ relative to SPST (figure 4*c*). Thus, outbreak sizes are amplified in the SPDT model. The initial disease reproduction abilities (figure 4*d*), which are calculated at the first simulation day, shows how the contact networks become favourable for diffusion with increasing $r_t$ in both SPDT and SPST models. The initial disease reproduction abilities in SPDT model are strongly influenced by $r_t$.

The temporal variation of disease prevalence for $r_t = \{15, 30, 45, 60\}$ min are presented in figure 5 while figure 6 shows the variations in disease reproduction rates $R_t$. The results show that the diffusion dynamics are strongly governed by $r_t$. The SDT network shows growing of disease prevalence $I_p$ from the initial 500 infected individuals for values of $r_t \geq 45$ min while $I_p$ drops for all other values (figure 5*b*). In the heterogeneous contact networks, the disease gradually reaches to the higher degree individuals, who have high contact rates, and hence the value of $R_t$ gradually increases [36,38]. This growth of $R_t$ is faster at high $r_t$ due to strong underlying connectivity (figure 6*b*). However, individuals with a high degree get infected earlier and the number of susceptible individuals reduces through time (see supplementary material [39]). Hence, an infection resistance force grows in the network and the rate $R_t$ of infected individuals decreases. Therefore, an initial

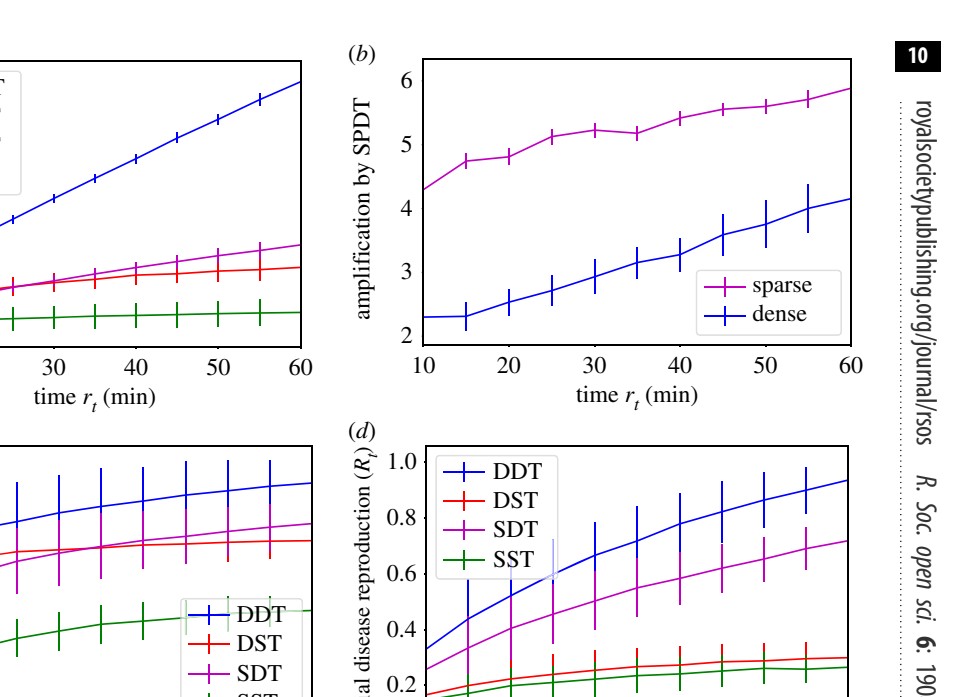

**Figure 4.** Diffusion dynamics including interquartile ranges on SPST and SPDT networks with particles decay rates $r_t$: (a) outbreak sizes—total number of infections, (b) amplification by SPDT model, (c) effective disease reproduction number $R_e$ and (d) initial disease reproduction number $R_t$.

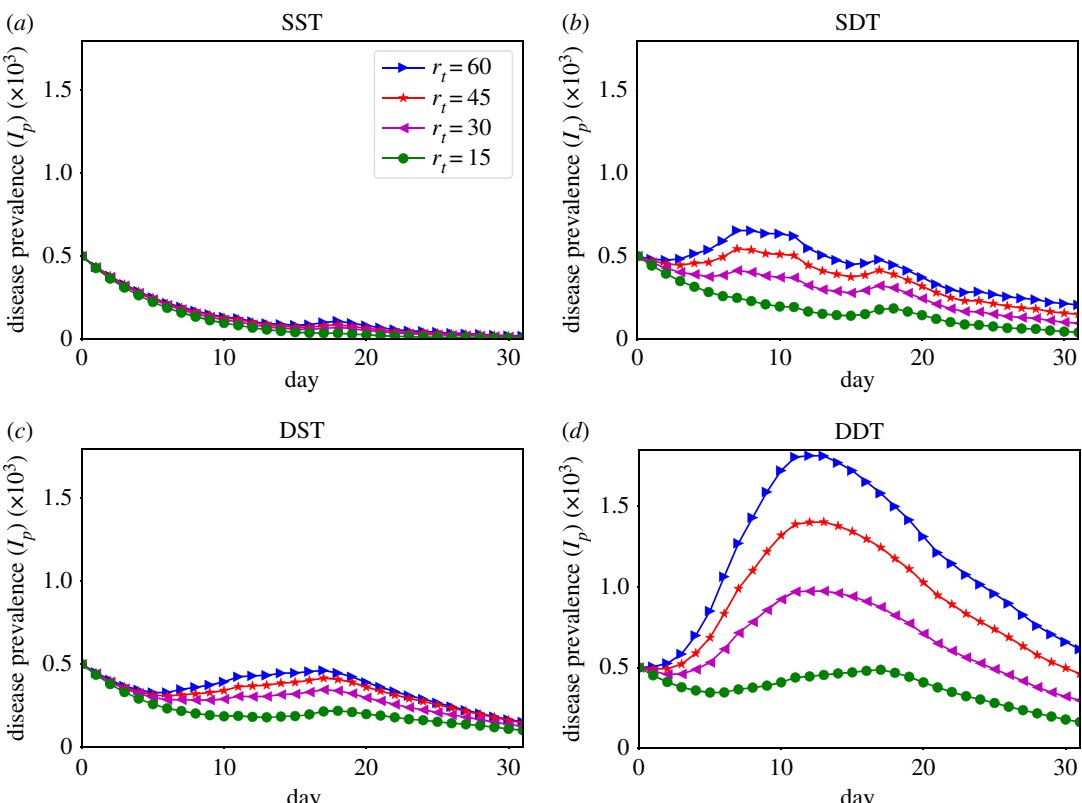

**Figure 5.** Disease prevalence, current number of infected individuals in the network, dynamics over simulation days for SPDT and SPST models with various particle decay rates $r_t$: (a) sparse SPST network (SST), (b) sparse SPDT network (SDT), (c) dense SPST network (DST) and (d) dense SPDT network (DDT).

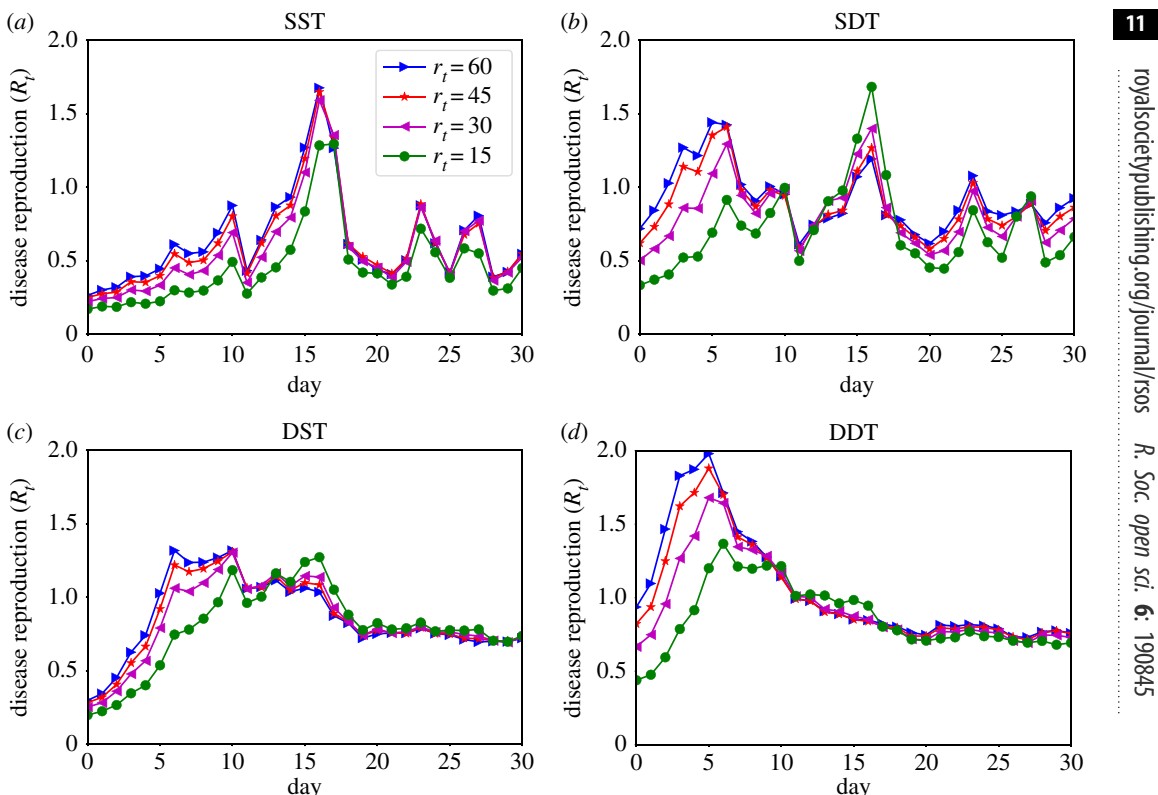

**Figure 6.** Daily variations in disease reproduction rate $R_t$ with particle decay rates $r_t$: (a) sparse SPST network (SST), (b) sparse SPDT network (SDT), (c) dense SPST network (DST) and (d) dense SPDT network (DDT).

small $R_t$ for $r_t \geq 45$ min quickly increases above one which grows $I_p$ as long as $R_t$ remains above one and then decreases (figure 6b). For $r_t < 45$ min, $R_t$ slowly grows above one due to the weak underlying network connectivity with low $r_t$ and $I_p$ decreases significantly with time. As a result, $I_p$ increases slightly and then starts dropping. In the SST network, $I_p$ could not grow for any value of $r_t$ due to very small initial $R_t$ and lack of connectivity for considering only direct links (figures 5a and 6a). The SDT network at $r_t = 10$ min shows a similar trend to the SST network (figure 6a,b). This is because the SDT network becomes similar to the SST network due to weak underlying connectivity at this low $r_t$ where creations of indirect links are limited.

The impact of the SPDT model becomes stronger in the dense DDT network: infected individuals apply their full infection potential by being every day in the network. The total infections (outbreak sizes) and the disease prevalence $I_p$ increase significantly (figures 4a and 5d). The DDT network is capable of increasing $I_p$ even at lower $r_t \geq 20$ min. Due to high link density, the disease reproduction rate $R_t$ in the DDT network reaches one quickly and then increases faster as time goes (figure 6d). The rate $R_t$ has multiple effects on the disease prevalence $I_p$. For the higher value of $I_p$ and $R_t > 1$, small changes in $R_t$ have a large increase in $I_p$. Small variations in $r_t$ change $R_t$ which in turn significantly changes $I_p$ in the DDT network. Conversely, $I_p$ first drops for all values of $r_t$ in the DST network and then starts increasing after some days as high degree individuals are infected [39]. However, this increase is not within the same range as the DDT network due to a weak $R_t$ and a lack of underlying connectivity. Similarly, the DDT network with low $r_t < 20$ min behaves comparatively to the DST network as the underlying connectivity becomes weak.

## 4.3. Diffusion for various disease parameters

The impact of particle decay rates $r_t$ on SPDT diffusion dynamic increases with increasing infectiousness $\sigma$. To analyse this, we run simulations for $\sigma = \{0.33, 0.4, 0.5\}$ on both SPST and SPDT networks with varying $r_t$. The results are presented in figure 7a,b. The amplification by SPDT model increases as $\sigma$ increases (figure 7a). The required $r_t$ to grow the disease prevalence $I_p$ in the SPDT model reduces with increasing $\sigma$ (see electronic supplementary material [39]). In addition, we note that the SPDT model makes a longer linear amplification (continuous growth of infection amplification with increasing $r_t$, which should stop at some

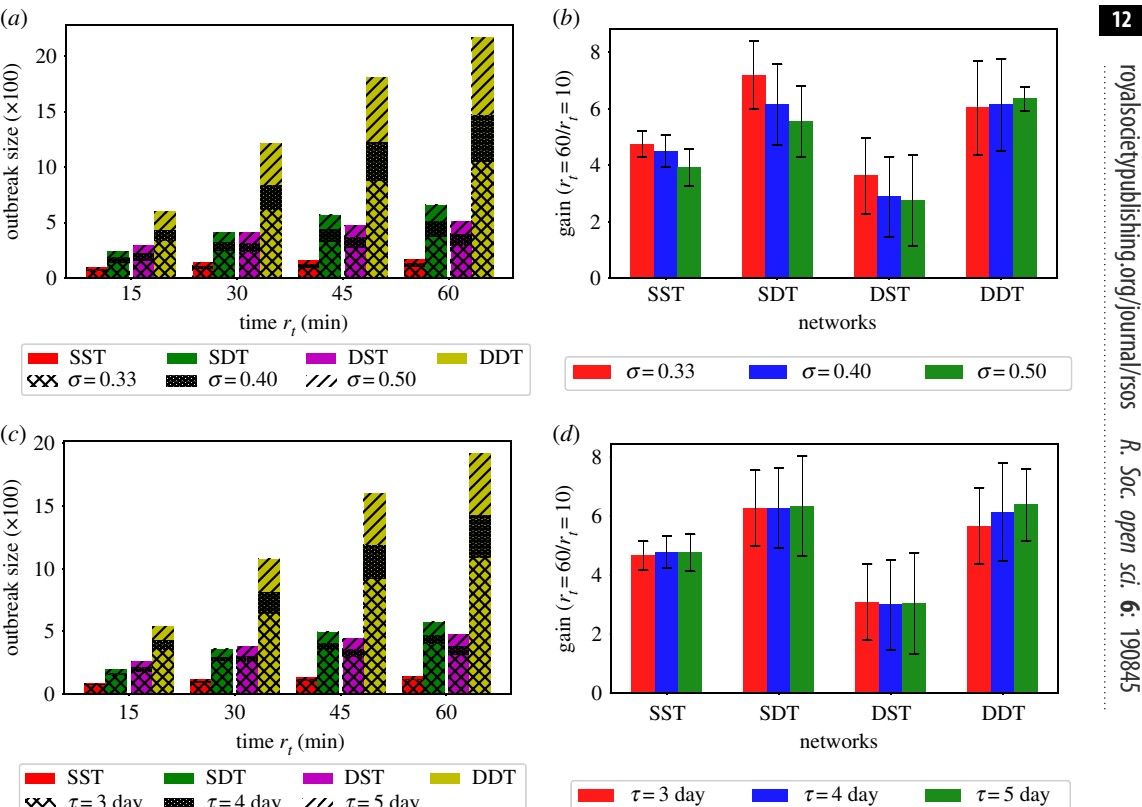

**Figure 7.** Diffusion dynamics for various disease parameters in the different networks: (*a*) outbreak sizes for various infectiousness $\sigma$-different pattern showing amplification for increasing $\sigma$, (*b*) gains in outbreak sizes for changing $r_t$ from 10 min to 60 min for various $\sigma$, (*c*) outbreak sizes for various infectious period $\tau$-different pattern showing amplification for increasing $\tau$, and (*d*) gains in outbreak sizes for changing $r_t$ from 10 min to 60 min for various $\tau$.

value of $r_t$) in dense networks. Except for the DDT network, the growth in the total infection gain at low $r_t$ due to an increase in $\sigma$ is higher compared to that at high $r_t$. This is shown in the figure 7*b* through the ratios of total infections at $r_t = 60$ min and $r_t = 10$ min. However, the DDT network shows a different behaviour as the growth in total infection gain at high $r_t$ still increases with $\sigma$ increases. Having higher link density and more high degree individuals, the DDT network can achieve a stronger infection force at high $r_t$ to override the infection resistance force coming from the reduction in susceptible individuals. However, the other three networks are affected by the infection resistance force at high $r_t$ more strongly than the low $r_t$ as they lack underlying connectivity (the sparse SST and DST networks only account for indirect links and SDT network has low link density).

A longer infectious period $\tau$ also increases the disease reproduction ability $R_t$ of infected individuals as recovery forces from infection are reduced. The diffusion behaviours for $\bar{\tau} = \{3, 4, 5\}$ days with constant $\sigma = 0.33$ are shown in figure 7. This also increases total infection (figure 7*c*) and reduces $r_t$ to grow disease prevalence $I_p$ (see electronic supplementary material [39]). In this case, the DDT network supports longer linear amplification as well while for other networks it reaches a steady state. In addition, the delays to reach a peak $I_p$ become longer as $\bar{\tau}$ increases. This is because $R_t$ is maintained over one for longer which grows $I_p$ for longer [39]. As a result, the disease persists within the population for a longer time in the SPDT model than the SPST model.

## 4.4. New diffusion dynamics of SPDT

The SPDT model introduces novel diffusion behaviours that are not observed in the SPST diffusion model. The underlying network connectivity in the SPDT model is changed with the particle decay rates $r_t$. The impact of this property on diffusion dynamics are not reproducible by the SPST model. This is observed by reconstructing equivalent SPDT diffusion dynamics for $\sigma = 0.33$ on the corresponding SPST network with strong $\sigma$ (figure 8*a*,*b*). With $\sigma = 0.85$, the SPST diffusion dynamics

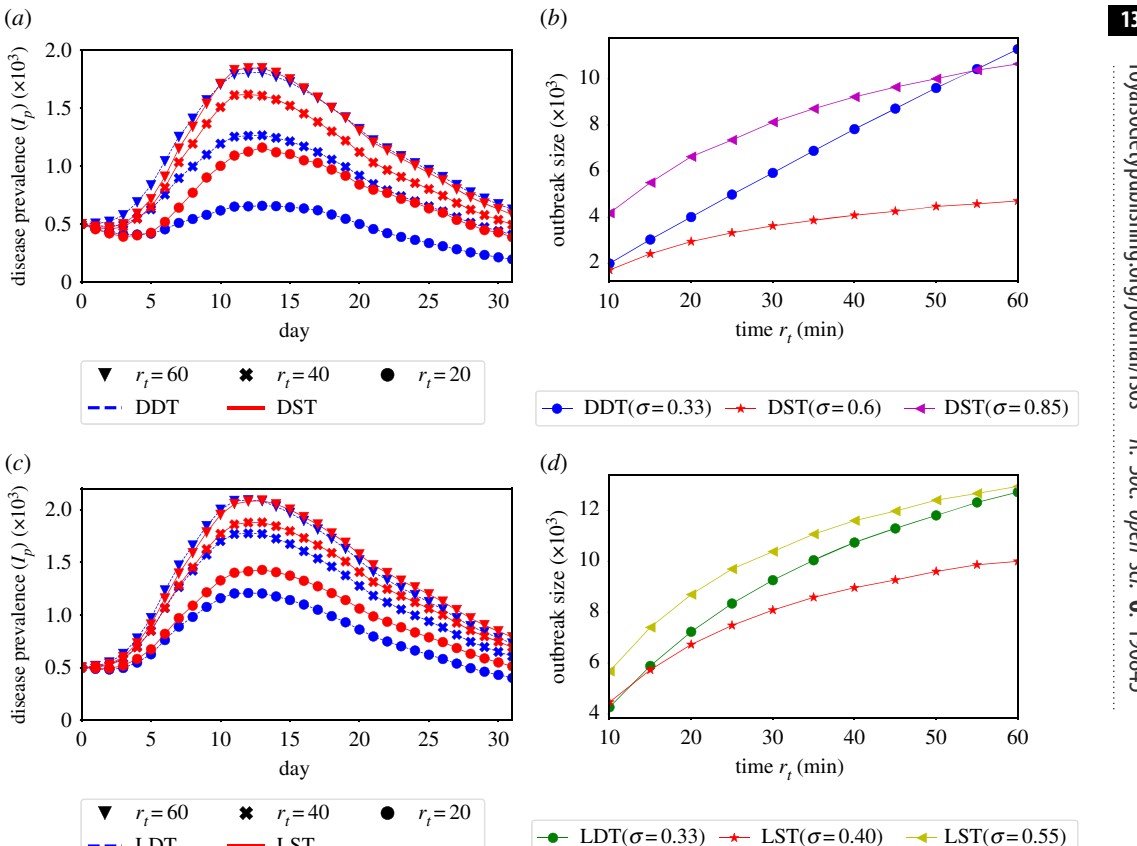

**Figure 8.** Reconstructing SPDT diffusion dynamics by the SPST model and their differences over various particle decay rates $r_t$: (a) comparing SPDT diffusion with $\sigma = 0.33$ and SPST diffusion with $\sigma = 0.85$, (b) comparing outbreak sizes of the SPDT model with $\sigma = 0.33$ and that of the SPST model with $\sigma = 0.60$ and $\sigma = 0.85$, (c) comparing LDT diffusion with $\sigma = 0.33$ and LST diffusion with $\sigma = 0.55$ and (d) comparing outbreak sizes of the LDT model with $\sigma = 0.33$ and that of the LST model with $\sigma = 0.55$ and $\sigma = 0.40$.

become similar to that of the SPDT model at $r_t = 60$ min (figure 8a) and outbreak sizes in both models are close to each other (figure 8b). However, diffusion dynamics and outbreak sizes in the SPST model are overestimated when $r_t$ is lowered at $\sigma = 0.85$. On the other hand, the SPST model with $\sigma = 0.6$ shows the same total infections of the SPDT model at $r_t = 10$ min but underestimates at the higher values of $r_t$. This is because the underlying connectivity in the SPST model does not vary with changing $r_t$ and spreading dynamic variations are limited.

We also find the indirect transmission links significantly increase diffusion dynamics even if the underlying connectivity in the SPST and SPDT model is made the same. This is observed from the diffusion dynamics on the LST and LDT networks which maintain the same number of active users (having links every day) and the same link densities. In addition, the LDT network can be described such that some direct links of the LST network are appended with indirect links to generate it. Therefore, the underlying connectivity is not changed with $r_t$, but the links get stronger with increasing $r_t$. The LDT network still shows significantly stronger disease prevalence relative to the LST network (figure 9a,b). By including indirect links, the LDT network achieves a strong disease reproduction rate $R_e$ and produces a high disease prevalence $I_p$.

Our experiments also show that indirect transmission links are more affected by $r_t$ compared to direct links. This phenomenon is also not reproducible by the SPST model which is observed reconstructing LDT diffusion dynamics on the LST network with increasing $\sigma$ (figure 8c,d). With $\sigma = 0.55$, the LST diffusion dynamics become similar to the LDT diffusion and causes the same number of infections at $r_t = 60$ min. However, the diffusion dynamics and outbreak sizes are overestimated at low $r_t$. By contrast, LST network with $\sigma = 0.4$ shows the same outbreak sizes as the LDT network at $r_t = 10$ min but underestimates at higher values of $r_t$. This difference is happening because the indirect link propensity is

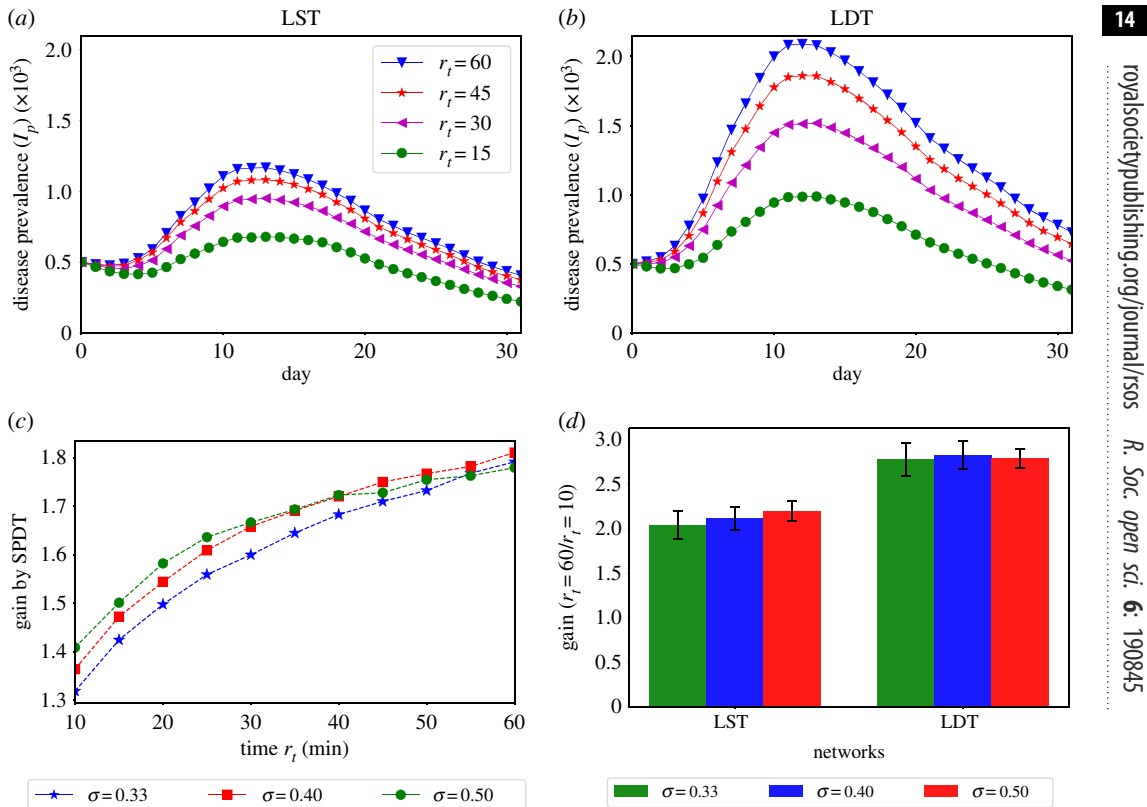

**Figure 9.** Diffusion on SPST and SPDT networks with controlled links densities: (*a*) diffusion dynamics on LST network at $\sigma = 0.33$ varying $r_t$, (*b*) diffusion dynamics on LDT network at $\sigma = 0.33$ varying $r_t$, (*c*) gain by the SPDT model over the SPST model (LDT/LST) at various $\sigma$ when both have same link densities and (*d*) infection gains for changing $r_t = 10$ min to $r_t = 60$ min at various $\sigma$.

more sensitive to $r_t$ than direct links. However, the variations in the reconstructed LST diffusion are small compared to those of the DST network as the underlying connectivity is diminished in the LDT network.

The SPDT model assumes longer linear amplifications of infections with $r_t$. We have seen the growth in SPDT amplification increases at high $r_t$ as $\sigma$ increases while it drops in the SPST network (figure 7). The underlying cause is found studying diffusion on LST and LDT networks for $\sigma = \{0.33, 0.4, 0.5\}$ (figure 9*c*,*d*). Unlike the previous diffusion dynamics (figure 7), the gain across $r_t$ (total infections at $r_t = 60$/total infections at $r_t = 10$) drops slightly in the LDT network which is the behaviour of the SPST model where the growth in amplification at low $r_t$ is more than that at high $r_t$ when $\sigma$ increases. Due to its having strong $R_e$ for indirect links, the infection force in the LDT network overrides infection resistance force at $\sigma = 0.4$, but it is affected at $\sigma = 0.5$. The LDT network gain reaches a steady state which did not happen in the DDT network. Thus, the SPDT model with underlying network dynamics supports extended spreading opportunities.

# 5. Discussion

## 5.1. Diffusion on dynamic networks

Current diffusion models only consider concurrent interactions as a cause of individual-level transmission. However, there are several real scenarios where indirect delayed interactions can also cause transmission of infectious items and many infectious diseases spread this way [5,40,41]. Similar diffusion mechanisms occur for the dissemination of messages in social media and in ant colonies [14,16]. The importance of indirect transmission is studied in several infectious disease spreading models [30,42–44]. However, they did not account for the individual-level indirect transmissions in developing disease outbreaks. For airborne diseases, suspension of particles in the air and their risk to transmit disease are widely discussed, but how this suspension can contribute in diffusion dynamics within a population remains underexplored [12,19]. To our knowledge, the work of [16] has only

considered the individual-level indirect transmissions in diffusion phenomena. This model focused on environmental factors affecting indirect transmission of infectious agents and the corresponding diffusion dynamics. In the airborne disease spreading, however, we need to focus on individual movement behaviours as well as their susceptibility and infectivity to the infectious agent. Moreover, this study did not consider how the network properties and their impacts are changed by including indirect transmission.

In our study, we have introduced a SPDT diffusion model accounting for the indirect transmissions along with the direct transmissions. The SPDT model is integrated with a risk assessment method that explicitly captures the individual movements through different timing parameters. This model is capable of including individual heterogeneity for susceptibility and infectivity. Random assignment of particle removal rates can emulate the variation in the environmental conditions and their impact on diffusion dynamics.

Inclusion of indirect transmissions changes the network topology: (1) strengthening the existing direct links by appending indirect links and (2) connecting individuals who are not connected with the direct links. Therefore, the underlying connectivity gets denser and stronger in the SPDT model compared to the SPST model and diffusion dynamics are amplified. However, the enhancement of SPDT diffusion is varied with particle decay rates $r_t$, which define the links infectivity, and disease parameters. The strong underlying connectivity increases reachability among individuals [23,36,39]. Hence, the disease reaches the high degree individuals faster as $r_t$ increases. Thus, there is a threshold value of $r_t$ to grow disease prevalence $I_p$ in the SPDT model. The threshold $r_t$ reduces in the networks with high link density and for strong disease parameters.

The SPDT model can capture realistic interactions for some diseases producing effective disease reproduction number $R_e$ greater than one and making significant outbreaks, while the SPST model could not produce $R_e$ at any conditions. This shows the SPDT model based on indirect links is more realistic than the SPST model based on the direct links. The study of [45] on the real influenza outbreaks in various regions shows that the effective disease reproduction number will be in the range [1.06-3.0]. The SPDT model attains realistic disease reproduction number $R_e$ having some values in this range at some values of $r_t$, while the SPST model could not achieve realistic $R_e$ for any $r_t$ in our simulations. This quantitative assessment also shows the ability of the SPDT model to capture realistic disease dynamics. The SPDT diffusion model can be applied to study influenza seasonality through modelling the particle decay rates $r_t$ which can be varied over time with the fluctuations in the weather factors such as humidity and temperature [46,47].

## 5.2. Limitations

The applied individual contact networks are built among the Momo users and the contacts of Momo users with other individuals are not included in our simulations. In addition, the networks do not include the contacts, while users are not using the app. Therefore, some potential transmission links could be missed and outbreak sizes could be different. With the known sparse contact data, we reconstructed the denser versions of empirical network repeating the available links for the missing days. This process has the following limitations: (1) limited link variation. When a user is present in the network for one day with only one indirect link, this indirect link is repeated for the missing 31 other days. This can not capture link variations for this user in reality, and (2) limited contact neighbours. When a user only presents one day, the same links are repeated for all missing days with the same old neighbours and new neighbours are not included. Thus, the disease spreading may be underestimated due to these limitations. Despite these issues, the Momo dataset provides a city-scale population dataset with high position resolution (compared to cellular call data records [41,48] for instance), and therefore capture population movements and contacts at the sufficiently accurate resolution to drive useful observations on the spreading dynamics.

We have used a simplified infection risk assessment model where links between two individuals are created if they are within 20 m of each other based on the GPS locations. However, the collected GPS locations may have errors when phones use Cell-ID or Wi-Fi signal for estimating GPS locations [49]. Thus, the individuals beyond 20 m distance might have created links in our developed networks and overestimation can occur. In the calculation of exposures, particle concentration decay over distance from an infected individual is neglected. The infection risk model is assigned random particle decay rates picking up from a range to mimic the heterogeneity of interaction locations due to architecture, wind-flow, humidity and temperate variations. However, this variation is not fitted with real-world data. We also avoid the heterogeneity of individuals in infectious particle generation, particle inhalation and susceptibility to disease.

## 5.3. Future works

The SPDT diffusion dynamics are highly influenced by the individual-level interaction data that are difficult to obtain from real scenarios. Thus, it would be interesting to develop synthetic networks capturing the properties of real contact traces. In this study, we have only studied the overall diffusion dynamics on the contact networks. The indirect transmission links may change local contact structure and hence disease emergence conditions can be altered in the SPDT model. This would be an interesting research direction to assess the potential of indirect links in emerging diseases. As the inclusion of indirect links makes one individual connect with others, the probability of being super-spreader increases. Thus, an important research direction is to find the vaccination strategies for SPDT model. The individual-level contact mechanisms also influence the higher order network properties. It would be interesting to know how the modified higher order networks properties with indirect links influence the diffusion dynamics. Contact generalizations and finding the relationship between them and the total epidemic size is an interesting research direction for future work. We have assumed that the infectious particles can travel up to 20 m in horizontal distance. It would be interesting to know to what extent the diffusion dynamics are varied in our model compared to metapopulation modelling approaches, in which locations (households, schools, communities) are captured with systems of equations and the underlying network structure describes movements of contacts between communities.

Data accessibility. Data and relevant code for this research work are stored in the Open Github repository and can be accessed at https://github.com/mszamalbd/Real-SPDT-Contact-Networks and have been archived within the Zenodo repository: https://doi.org/10.5281/zenodo.3332816 [31].

Funding. This study was supported by the Australian Research CouncilGrant (ARC - DP170102794) and Commonwealth Scientific and Industrial Research Organisation (CSIRO). Md.S. was partially supported by the CSIRO and ARC-DP170102794. B.M. was partially supported by the ARC-DP170102794. R.J. and F.d.H. were supported by the CSIRO.

Acknowledgements. The authors gratefully acknowledge the Distributed Sensing System Group, Data61, CSIRO for providing research facilities for this research. We are also thankful for the Principal Research Scientist Dean Paini, Postdoctoral Fellow Jessica Liebig and Associate Professor Lauren Gardner for their comments and suggestions on this research.

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
