## [Reviewer comments · Royal Society Open Science]

Review History

RSOS-190845.R0 (Original submission)

Review form: Reviewer 1

Is the manuscript scientifically sound in its present form?

Yes

Are the interpretations and conclusions justified by the results?

Yes

Is the language acceptable?

Yes

Is it clear how to access all supporting data?

Yes

Do you have any ethical concerns with this paper?

No

Have you any concerns about statistical analyses in this paper?

No

Recommendation?

Major revision is needed (please make suggestions in comments)

Comments to the Author(s)

This study explores the importance of considering indirect directions while modeling airborne transmitting infections through contact networks. The contact network is realistic as it is based on data from social media app momo. The study finds that the inclusion of indirect links increases disease diffusion through contact networks. Although these results are obvious, I find the results on LST and LDT networks particularly interesting since they indicate that the impact of indirect links on disease diffusion is more complex than just increasing network's link density.

I enjoyed reading the manuscript and think it makes an important contribution to the literature. I have a few comments to help improve the clarity of the results presented in the study.

Abstract:

- (1) I suggest mentioning the exciting result that indirect links increase diffusion even if underlying connectivity in the SPST and SPDT network is the same.
- (2) I also agree with previous reviewers that abstract could focus on how (a) SPDT models are more realistic for airborne disease and/or (b) diffusion dynamics of an airborne disease cannot be reproduced by the current SPST models.
- (3) Please note that clarifying the superiority of SPDT over SPST for airborne transmission is important. The current phrasing makes it seem that the SPST model is inadequate for all infectious diseases which is certainly not true. For example, the SPST model is adequate for sexually transmitted disease, infectious diseases that spread by physical contact, etc.
- (4) What does increasing the dynamics of disease imply? Do the authors mean increase disease "diffusion rate" instead?
- (5) "networks with low link densities". Unless I misunderstood, this statement is referring to the sparse SPDT/SPST networks. Based on the results, I don't see how sparse SPDT model "particularly" increases disease diffusion. For instance, results (page 10 line 50) states that "the impact of SPDT model becomes stronger in the dense DDT network). The impact of DDT networks is also much higher in Figure 7 compared with SDT networks. Please clarify.

Introduction:

- (6) It would be useful to discuss how the current model formulation conceptually differs from previous SPDT model in the literature by Richardson and Goroehowski (2015).
- (7) Page 5, line 38-39: How exactly were the links reconstructed for missing days? By repeating the same links as the last available day with movement data, or by taking an "average" link behavior over all days where the data were available?

Methods:

- (8) How is amplification calculated? It does not appear to be simply = (disease outcome for SPDT models/ disease outcome for SPST model) because in that case amplification by dense SPDT model will be more than sparse SPDT model (Figure 4A)? Please clarify and mention the calculations in the main text.

Results:

(9) Page 9, line 57-58: I wouldn't consider R_e for dense SPDT model to be strong compared to dense SPST model, because the interquartile ranges of DST and DDT overlap for all values of r_t in Figure 4C. Is there perhaps another explanation for why outbreak sizes are amplified up to 4.3 times in the dense networks?

Discussion:

(10) It's strange to see new results (Figures 8 and 9) being introduced in the discussion section. Please move these to the results section.

Minor comments:

(1) Page 5, Line 7: "Contrasting disease". Shouldn't it be "contracting a disease"?

(2) Page 6, line 33-34: It is mentioned here that sparse SPST and SPDT networks will be denoted as SST and SDT, yet the acronym usage is reverted back to SPST and SPDT in the same paragraph and rest of the paper. Please be consistent in the usage of acronyms. If there is a difference between SPST(SPDT) and SST(SPST), please clearly state the distinction.

(3) Page 7, line 45-46: "continuous reduction dies out the disease". It is unclear what this means. Please rephrase.

(4) Page 8, line 39: the interaction \square an interaction

(5) Page 8, line 46: remove period after r_t

(6) Page 8, line 47: It is unclear what "the same changes" mean with regards to the previous statement.

(7) Figure 3 legend: spst and spdt should be in upper case. Also, these are referred to as SST and SDT networks in the text. Please be consistent with the acronym usage.

(8) Figure 3C and D: What does the solid red line labeled as spdt represent? Where is it in the figure?

(9) Figure 2 and 3: It will be easier to follow these figures if the same coloring scheme for r_t values is followed for all figures.

(10) Figure 4: Please change the y-axis label of 4A to "outbreak size" so that it is consistent with the rest of the figures. "Total number of infections" can be mentioned in brackets as a definition.

Review form: Reviewer 2

Is the manuscript scientifically sound in its present form?

Yes

Are the interpretations and conclusions justified by the results?

Yes

Is the language acceptable?

Yes

Is it clear how to access all supporting data?

Yes

Do you have any ethical concerns with this paper?

No

Have you any concerns about statistical analyses in this paper?

No

Recommendation?

Accept with minor revision (please list in comments)

Comments to the Author(s)

Below are minor comments.

Abstract:

I much preferred the previous abstract. It was easier to read and had a nice flow to it. This abstract needs cleaning up. i.e.

“Most of the current diffusion models only consider direct interactions among individuals to build underlying infectious items transmission networks.”

1) I think it was fine to refer to these other models as “network models”.

2) “to build underlying infectious items transmission networks” is not clear. I think what you are trying to say (which I agree with) is that ‘Most network models only consider direct contacts capable of transmitting disease’

P 2. L41-52. There are a lot of abbreviations introduced, but not defined. I feel like much of this should be in the methods section.

P 3 Line 16, Fig1 : define ‘L’ in the text (as the blue circle) and in the figure. Or leave it out completely.

P 3. 57. Change ‘directed transmission’ to ‘direct transmission’

P 4. L7 “we integrate...” This sentence needs clarifying.

P4. 7-14. You use ‘proximity’ and ‘location L’ to represent the same think. Just use one.

P4. L42. change ‘an SPDT links’ to ‘SPDT links’. Make this change elsewhere as well (ie pg 5 L20, ‘an SPDT to a SPDT’).

P4. (2.8) Does ‘E’, overall exposure, include both direct and indirect links? As it is written, it looks like it only includes indirect links. Please clarify. Should E not include both E_d and E_l?

P4. L59. This is the first time you have mentioned virus. Until this point you have kept the infectious agent general. I would suggest continuing to do so, or state that you are modelling viral dynamics earlier.

P5 L24-26. "When the host user's last..." This is an important sentence and needs cleaning up/clarification.

P5 L 26-17. Perhaps change "of last update" to "for up to 200 min since the last update" (?)

P5. L 49-50. 'diffusion behaviours of SPDT model comparing to SPST model'. change to: 'of the SPDT model' or 'of SPDT models'... etc.

P6. Under Characterizing Metrics. The first sentence needs clarifying. I would not refer to I_n and I_p as parameters. Also, what is the symbol for individuals recovered? Is this I_r (which you use below?). Do the subscripts 'n' and 'p' implicitly represent time (simulation day)? Also can you elaborate on how you take the average of R_t to get R_e .

P7. L11. Is ' r_t ' the same as ' r ' defined above. Is the ' t ' subscript indicative of decay time? I think you may have confused decay rate with the lifespan of a particle. I would think that as particles decay more quickly (ie the largest rate $r_t=60$), you would have fewer particles lingering about, since they are decaying quickly. I am assuming that r_t is the lifespan, and r is the rate. Make this clean and relabel the figure.

P9 L10-12. "...reaches to the individuals,..."
change to "reaches the higher degree individuals"

P9 L55 subscript on R missing.

P11 consider using different symbol for recovery and infectivity. You use σ for both. (see top of P11 and (3.1). This becomes more confusing later on in your discussion on P14.

Decision letter (RSOS-190845.R0)

21-Jun-2019

Dear Mr Shahzamal,

On behalf of the Editors, I am pleased to inform you that your Manuscript RSOS-190845 entitled "Indirect interactions influence contact network structure and diffusion dynamics" has been accepted for publication in Royal Society Open Science subject to minor revision in accordance with the referee suggestions. Please find the referees' comments at the end of this email.

The reviewers and handling editors have recommended publication, but also suggest some minor revisions to your manuscript. Therefore, I invite you to respond to the comments and revise your manuscript.

- Ethics statement

- Data accessibility

<http://datadryad.org/submit?journalID=RSOS&manu=RSOS-190845>

- Competing interests

- Authors' contributions

- Acknowledgements

- Funding statement

Because the schedule for publication is very tight, it is a condition of publication that you submit the revised version of your manuscript before 30-Jun-2019. Please note that the revision deadline will expire at 00.00am on this date. If you do not think you will be able to meet this date please let me know immediately.

If your manuscript is newly submitted and subsequently accepted for publication, you will be asked to pay the article processing charge, unless you request a waiver and this is approved by Royal Society Publishing. You can find out more about the charges at

<http://rsos.royalsocietypublishing.org/page/charges>. Should you have any queries, please contact openscience@royalsociety.org.

Kind regards,

on behalf of Dr Francois Fages (Associate Editor) and Marta Kwiatkowska (Subject Editor)
openscience@royalsociety.org

Associate Editor Comments to Author (Dr Francois Fages):

Dear Authors

It is my pleasure to accept your article with minor revisions. Please do take into consideration all comments of the peer reviews below for preparing the final version of your manuscript.

Best regards

Reviewer comments to Author:

Reviewer 1:

This study explores the importance of considering indirect directions while modeling airborne transmitting infections through contact networks. The contact network is realistic as it is based on data from social media app momo. The study finds that the inclusion of indirect links increases disease diffusion through contact networks. Although these results are obvious, I find the results on LST and LDT networks particularly interesting since they indicate that the impact of indirect links on disease diffusion is more complex than just increasing network's link density.

I enjoyed reading the manuscript and think it makes an important contribution to the literature. I have a few comments to help improve the clarity of the results presented in the study.

Abstract:

- (1) I suggest mentioning the exciting result that indirect links increase diffusion even if underlying connectivity in the SPST and SPDT network is the same.
- (2) I also agree with previous reviewers that abstract could focus on how (a) SPDT models are more realistic for airborne disease and/or (b) diffusion dynamics of an airborne disease cannot be reproduced by the current SPST models.
- (3) Please note that clarifying the superiority of SPDT over SPST for airborne transmission is important. The current phrasing makes it seem that the SPST model is inadequate for all infectious diseases which is certainly not true. For example, the SPST model is adequate for sexually transmitted disease, infectious diseases that spread by physical contact, etc.

(4) What does increasing the dynamics of disease imply? Do the authors mean increase disease “diffusion rate” instead?

(5) “networks with low link densities”. Unless I misunderstood, this statement is referring to the sparse SPDT/SPST networks. Based on the results, I don’t see how sparse SPDT model “particularly” increases disease diffusion. For instance, results (page 10 line 50) states that “the impact of SPDT model becomes stronger in the dense DDT network). The impact of DDT networks is also much higher in Figure 7 compared with SDT networks. Please clarify.

Introduction:

(6) It would be useful to discuss how the current model formulation conceptually differs from previous SPDT model in the literature by Richardson and Goroehowski (2015).

(7) Page 5, line 38-39: How exactly were the links reconstructed for missing days? By repeating the same links as the last available day with movement data, or by taking an “average” link behavior over all days where the data were available?

Methods:

(8) How is amplification calculated? It does not appear to be simply = (disease outcome for SPDT models/ disease outcome for SPST model) because in that case amplification by dense SPDT model will be more than sparse SPDT model (Figure 4A)? Please clarify and mention the calculations in the main text.

Results:

(9) Page 9, line 57-58: I wouldn’t consider R_e for dense SPDT model to be strong compared to dense SPST model, because the interquartile ranges of DST and DDT overlap for all values of r_t in Figure 4C. Is there perhaps another explanation for why outbreak sizes are amplified up to 4.3 times in the dense networks?

Discussion:

(10) It’s strange to see new results (Figures 8 and 9) being introduced in the discussion section. Please move these to the results section.

Minor comments:

(1) Page 5, Line 7: “Contrasting disease”. Shouldn’t it be “contracting a disease”?

(2) Page 6, line 33-34: It is mentioned here that sparse SPST and SPDT networks will be denoted as SST and SDT, yet the acronym usage is reverted back to SPST and SPDT in the same paragraph and rest of the paper. Please be consistent in the usage of acronyms. If there is a difference between SPST(SPDT) and SST(SPST), please clearly state the distinction.

(3) Page 7, line 45-46: “continuous reduction dies out the disease”. It is unclear what this means. Please rephrase.

(4) Page 8, line 39: the interaction □ an interaction

(5) Page 8, line 46: remove period after r_t

(6) Page 8, line 47: It is unclear what “the same changes” mean with regards to the previous statement.

(7) Figure 3 legend: spst and spdt should be in upper case. Also, these are referred to as SST and SDT networks in the text. Please be consistent with the acronym usage.

(8) Figure 3C and D: What does the solid red line labeled as spdt represent? Where is it in the figure?

(9) Figure 2 and 3: It will be easier to follow these figures if the same coloring scheme for r_t values is followed for all figures.

(10) Figure 4: Please change the y-axis label of 4A to “outbreak size” so that it is consistent with the rest of the figures. “Total number of infections” can be mentioned in brackets as a definition.

Reviewer 2:

Below are minor comments.

Abstract:

I much preferred the previous abstract. It was easier to read and had a nice flow to it. This abstract needs cleaning up. i.e.

“Most of the current diffusion models only consider direct interactions among individuals to build underlying infectious items transmission networks.”

1) I think it was fine to refer to these other models as “network models”.

2) “to build underlying infectious items transmission networks” is not clear. I think what you are trying to say (which I agree with) is that ‘Most network models only consider direct contacts capable of transmitting disease’

P 2. L41-52. There are a lot of abbreviations introduced, but not defined. I feel like much of this should be in the methods section.

P 3 Line 16, Fig1 : define ‘L’ in the text (as the blue circle) and in the figure. Or leave it out completely.

P 3. 57. Change ‘directed transmission’ to ‘direct transmission’

P 4. L7 “we integrate...” This sentence needs clarifying.

P4. 7-14. You use ‘proximity’ and ‘location L’ to represent the same think. Just use one.

P4. L42. change ‘an SPDT links’ to ‘SPDT links’. Make this change elsewhere as well (ie pg 5 L20, ‘an SPDT to a SPDT’).

P4. (2.8) Does ‘E’, overall exposure, include both direct and indirect links? As it is written, it looks like it only includes indirect links. Please clarify. Should E not include both E_d and E_I ?

P4. L59. This is the first time you have mentioned virus. Until this point you have kept the infectious agent general. I would suggest continuing to do so, or state that you are modelling viral dynamics earlier.

P5 L24-26. "When the host user's last..." This is an important sentence and needs cleaning up/clarification.

P5 L 26-17. Perhaps change "of last update" to "for up to 200 min since the last update" (?)

P5. L 49-50. 'diffusion behaviours of SPDT model comparing to SPST model'. change to: 'of the SPDT model' or 'of SPDT models' ... etc.

P6. Under Characterizing Metrics. The first sentence needs clarifying. I would not refer to I_n and I_p as parameters. Also, what is the symbol for individuals recovered? Is this I_r (which you use below?). Do the subscripts 'n' and 'p' implicitly represent time (simulation day)? Also can you elaborate on how you take the average of R_t to get R_e .

P7. L11. Is ' r_t ' the same as ' r ' defined above. Is the ' t ' subscript indicative of decay time? I think you may have confused decay rate with the lifespan of a particle. I would think that as particles decay more quickly (ie the largest rate $r_t=60$), you would have fewer particles lingering about, since they are decaying quickly. I am assuming that r_t is the lifespan, and r is the rate. Make this clean and relabel the figure.

P9 L10-12. "...reaches to the individuals,..."
change to "reaches the higher degree individuals"

P9 L55 subscript on R missing.

P11 consider using different symbol for recovery and infectivity. You use σ for both. (see top of P11 and (3.1). This becomes more confusing later on in your discussion on P14.

Author's Response to Decision Letter for (RSOS-190845.R0)

See Appendix A.

Decision letter (RSOS-190845.R1)

17-Jul-2019

Dear Mr Shahzamal,

I am pleased to inform you that your manuscript entitled "Indirect interactions influence contact network structure and diffusion dynamics" is now accepted for publication in Royal Society Open Science.

Kind regards,

on behalf of Dr Francois Fages (Associate Editor) and Marta Kwiatkowska (Subject Editor)
openscience@royalsociety.org

Associate Editor Comments to Author (Dr Francois Fages):

Dear Authors

It is my pleasure to accept your paper as is. You have correctly addressed the concerns of the reviewers in this revised version.

Congratulations.

Follow Royal Society Publishing on Twitter: [@RSocPublishing](https://twitter.com/RSocPublishing)
Follow Royal Society Publishing on Facebook:
<https://www.facebook.com/RoyalSocietyPublishing.FanPage/>
Read Royal Society Publishing's blog: <https://blogs.royalsociety.org/publishing/>

Appendix A

Indirect interactions influence contact network structure and diffusion dynamics

Md Shahzamal, Raja Jurdak, Bernard Mans, Frank de Hoog

Responses to reviewer's comments

Reviewer 1:

Comments to the Author:

This study explores the importance of considering indirect interactions while modeling airborne transmitting infections through contact networks. The contact network is realistic as it is based on data from social media app momo. The study finds that the inclusion of indirect links increases disease diffusion through contact networks. Although these results are obvious, I find the results on LST and LDT networks particularly interesting since they indicate that the impact of indirect links on disease diffusion is more complex than just increasing network's link density.

I enjoyed reading the manuscript and think it makes an important contribution to the literature. I have a few comments to help improve the clarity of the results presented in the study.

Abstract:

(1) I suggest mentioning the exciting result that indirect links increase diffusion even if underlying connectivity in the SPST and SPDT network is the same.

(2) I also agree with previous reviewers that abstract could focus on how (a) SPDT models are more realistic for airborne disease and/or (b) diffusion dynamics of an airborne disease cannot be reproduced by the current SPST models.

(3) Please note that clarifying the superiority of SPDT over SPST for airborne transmission is important. The current phrasing makes it seem that the SPST model is inadequate for all infectious diseases which is certainly not true. For example, the SPST model is adequate for sexually transmitted disease, infectious diseases that spread by physical contact, etc.

(4) What does increasing the dynamics of disease imply? Do the authors mean increase disease "diffusion rate" instead?

(5) "networks with low link densities". Unless I misunderstood, this statement is referring to the sparse SPDT/SPST networks. Based on the results, I don't see how sparse SPDT model "particularly" increases disease diffusion. For instance, results (page 10 line 50) states that "the impact of SPDT model becomes stronger in the dense DDT network). The impact of DDT networks is also much higher in Figure 7 compared with SDT networks. Please clarify.

Introduction:

(6) It would be useful to discuss how the current model formulation conceptually differs from previous SPDT model in the literature by Richardson and Gorochoowski (2015).

(7) Page 5, line 38-39: How exactly were the links reconstructed for missing days? By repeating the same links as the last available day with movement data, or by taking an “average” link behavior over all days where the data were available?

Methods:

(8) How is amplification calculated? It does not appear to be simply = (disease outcome for SPDT models/ disease outcome for SPST model) because in that case amplification by dense SPDT model will be more than sparse SPDT model (Figure 4A)? Please clarify and mention the calculations in the main text.

Results:

(9) Page 9, line 57-58: I wouldn't consider R_e for dense SPDT model to be strong compared to dense SPST model, because the interquartile ranges of DST and DDT overlap for all values of r_t in Figure 4C. Is there perhaps another explanation for why outbreak sizes are amplified up to 4.3 times in the dense networks?

Discussion:

(10) It's strange to see new results (Figures 8 and 9) being introduced in the discussion section. Please move these to the results section.

Minor comments:

(1) Page 5, Line 7: “Contrasting disease”. Shouldn't it be “contracting a disease”?

(2) Page 6, line 33-34: It is mentioned here that sparse SPST and SPDT networks will be denoted as SST and SDT, yet the acronym usage is reverted back to SPST and SPDT in the same paragraph and rest of the paper. Please be consistent in the usage of acronyms. If there is a difference between SPST(SPDT) and SST(SPST), please clearly state the distinction.

(3) Page 7, line 45-46: “continuous reduction dies out the disease”. It is unclear what this means. Please rephrase.

(4) Page 8, line 39: the interaction \diamond an interaction

(5) Page 8, line 46: remove period after r_t

(6) Page 8, line 47: It is unclear what “the same changes” mean with regards to the previous statement.

(7) Figure 3 legend: spst and spdt should be in upper case. Also, these are referred to as SST and SDT networks in the text. Please be consistent with the acronym usage.

(8) Figure 3C and D: What does the solid red line labeled as spdt represent? Where is it in the figure?

(9) Figure 2 and 3: It will be easier to follow these figures if the same coloring scheme for rt values is followed for all figures.

(10) Figure 4: Please change the y-axis label of 4A to “outbreak size” so that it is consistent with the rest of the figures. “Total number of infections” can be mentioned in brackets as a definition.

Response to the major comments of reviewer 1:

We are grateful and thank the reviewer 1 for spending time and making kind effort to review our manuscripts. The comments are very helpful to improve our manuscripts. We have explained our response to every comment based on our experiment results and relevant literature. The comments are marked by blue colour and response is started with ‘*Response*’ word. The relevant changes in the text of our manuscript are quoted here with ‘.....’.

Abstract:

Comment-1: I suggest mentioning the exciting result that indirect links increase diffusion even if underlying connectivity in the SPST and SPDT network is the same.

Response: Thanks to the reviewer 1 for noticing the important point of our results. We have included this important finding in the abstract of our manuscript. The relevant text is as follows:

‘Importantly, we also find that the diffusion dynamics including indirect links are not reproducible by the current SPST models, even if both SPDT and SPST networks assume the same underlying connectivity.’

Comment-2: I also agree with previous reviewers that abstract could focus on how (a) SPDT models are more realistic for airborne disease and/or (b) diffusion dynamics of an airborne disease cannot be reproduced by the current SPST models.

Response: This very important to reflect the finding of experiment. We would like to thank reviewer 1 for addressing these issues. We have revised our abstract and included the following arguments in the abstract.

‘By making the underlying connectivity denser and stronger due to the inclusion of indirect transmissions, SPDT models are more realistic than SPST models that are in current studies of various airborne diseases outbreaks.’

‘Importantly, we also find that the diffusion dynamics including indirect links are not reproducible by the current SPST models, even if both SPDT and SPST networks assume the same underlying connectivity. This is because the transmission dynamics of indirect links are different from that of direct links.’

Comment-3: Please note that clarifying the superiority of SPDT over SPST for airborne

transmission is important. The current phrasing makes it seem that the SPST model is inadequate for all infectious diseases which is certainly not true. For example, the SPST model is adequate for sexually transmitted disease, infectious diseases that spread by physical contact, etc.

Response: We have clarified this point rewriting our abstract as follows:

'By making the underlying connectivity denser and stronger due to the inclusion of indirect transmissions, SPDT models are more realistic than SPST models that are in current studies of various airborne diseases outbreaks.'

Comment-4: What does increasing the dynamics of disease imply? Do the authors mean increase disease "diffusion rate" instead?

Response: We have addressed this important comment and revised the text in abstract. Our modified text is as follows:

'The SPDT model significantly increases diffusion dynamics with a high rate of disease transmission.'

Comment-5: "networks with low link densities". Unless I misunderstood, this statement is referring to the sparse SPDT/SPST networks. Based on the results, I don't see how sparse SPDT model "particularly" increases disease diffusion. For instance, results (page 10 line 50) states that "the impact of SPDT model becomes stronger in the dense DDT network). The impact of DDT networks is also much higher in Figure 7 compared with SDT networks. Please clarify.

Response: As this is not a key result of our experiments, we have removed this point from the Abstract to avoid confusion.

Introduction:

Comment-6: It would be useful to discuss how the current model formulation conceptually differs from previous SPDT model in the literature by Richardson and Gorochowski (2015).

Response: We would like to thank reviewer 1 for suggesting to include clarification on how the proposed model differs from the previous SPDT model. We have clarified this point including the corresponding argument. We have included the following text in 1st paragraph of Section 5.1 on p.14.

'To our knowledge, the work of [16] has only considered the individual-level indirect transmissions in diffusion phenomena. This model focused on environmental factors affecting indirect transmission of infectious agents and the corresponding diffusion dynamics. In the airborne disease spreading, however, we need to focus on individual movement behaviours as well as their susceptibility and infectivity to the infectious agent. Moreover, this study did not consider how the network properties and their impacts are changed by including indirect transmission.'

Comment - 9 (Page 5, line 38-39): How exactly were the links reconstructed for missing days? By repeating the same links as the last available day with movement data, or by taking an “average” link behavior over all days where the data were available?

Response: This is the main mechanism to build our contact networks. Thanks to reviewer 1 for advising to add more explanation. In this process, all links from available days are picked up and placed to a random missing day. In this way, all missing days are filled up. This preserves the randomness of link creation in reality. The revised text is in 2nd paragraph in Section 3.2 of p.5 as follows:

‘In this process, all links of a day picked randomly from available days are copied to a random missing day.’

Methods:

Comment-8: How is amplification calculated? It does not appear to be simply = (disease outcome for SPDT models/ disease outcome for SPST model) because in that case amplification by dense SPDT model will be more than sparse SPDT model (Figure 4A)? Please clarify and mention the calculations in the main text.

Response: We have calculated the amplification for outbreak sizes. This is the ratio between the outbreak size in SPDT model and the outbreak size in SPST model. The revised text is in the 1st paragraph of Section 4.2 on p.8.

‘The amplification in outbreak size with the SPDT model is up to 5.6 times for sparse networks and 4.3 times for dense networks at $r_t=60$ min (Fig.4B).’

Results:

Comment-9 (Page 9, line 57-58): I wouldn’t consider R_e for dense SPDT model to be strong compared to dense SPST model, because the interquartile ranges of DST and DDT overlap for all values of r_t in Figure 4C. Is there perhaps another explanation for why outbreak sizes are amplified up to 4.3 times in the dense networks?

Response: We agree with Reviewer 1 that outbreak sizes are not only amplified in SPDT model due to R_e . Number of higher degree individuals and stronger underlying connectivity are the key factors to amplify the diffusion in SPDT model. The variation in initial R_t also influences the amplification in SPDT model. It is explained in the 2nd paragraph of Section 4.2 of our manuscript on p.9.

‘the disease gradually reaches to the higher degree individuals’

‘In the SPST network, I_p could not grow for any value of r_t due to very small initial R_t and lack of connectivity for considering only direct links’

Discussion:

Comment-10: It's strange to see new results (Figures 8 and 9) being introduced in the discussion section. Please move these to the results section.

Response: Agreed, we have moved it to the results section.

Response to the minor comments of Reviewer 1:

Comment-1 (Page 5, Line 7) : “Contrasting disease”. Shouldn't it be “contracting a disease”?

Response: We have replaced ‘Contrasting disease’ with ‘Contracting a disease’.

Comment-2 (Page 6, line 33-34): It is mentioned here that sparse SPST and SPDT networks will be denoted as SST and SDT, yet the acronym usage is reverted back to SPST and SPDT in the same paragraph and rest of the paper. Please be consistent in the usage of acronyms. If there is a difference between SPST(SPDT) and SST(SPST), please clearly state the distinction.

Response: We would like to thank Reviewer 1 for suggesting clarifying networks definition. SST, SDT, DST, DDT, LST and LDT networks are the different types of SPST and SPDT networks. In this paper, SPST network means network with direct links while SPDT networks mean network with both direct and indirect links. The SPST and SPDT networks are conceptual network definition. We have included an explanation in the 2nd paragraph of Section – 3.2 on p.5 of the manuscript.

“In this paper, Sparse SPDT network is denoted as SDT network and Sparse SPST network as SST network. However, SPDT network mean any network with both direct and indirect links while SPST network mean network with direct links only.”

Comment-3 (Page 7, line 45-46): “continuous reduction dies out the disease”. It is unclear what this means. Please rephrase.

Response: We have revised the text “continuous reduction dies out the disease” as “disease dies out if it continuously reduces” in Section 3.5 on p.6 of our manuscript.

Comment-4 (Page 8, line 39): the interaction \diamond an interaction

Response: We have updated our manuscript according to advice.

Comment-5 (Page 8, line 46): remove period after rt

Response: We have removed it.

Comment-6 (Page 8, line 47): It is unclear what “the same changes” mean with regards to the previous statement.

Response: Thanks to Reviewer 1 for noticing this. We have found that this is a redundant line added by mistaken. We have removed it.

Comment-7: Figure 3 legend: $spst$ and $spdt$ should be in upper case. Also, these are referred to as SST and SDT networks in the text. Please be consistent with the acronym usage.

Response: We have updated the legend and replaced the SST and SDT with SPST and SPDT. We want to discuss SPST and SPDT model in general. Thus, SST and SDT is removed.

Comment-8: Figure 3C and D: What does the solid red line labeled as $spdt$ represent? Where is it in the figure?

Response: We have added text in the caption what is meant by these. The included text in the caption of Figure 3 of our manuscript of p.8 is as follows:

‘SPST (dashed line) and SPDT (solid lines) networks properties: A) degree distribution in static networks, B) clustering co-efficient distribution in static networks, C) daily average degree in dynamic networks, and D) daily average clustering co-efficient in dynamic networks’

Comment-9: Figure 2 and 3: It will be easier to follow these figures if the same coloring scheme for rt values is followed for all figures.

Response: We have updated the colours for the lines to be consistent in Figures 2 and 3.

Comment-10: Figure 4: Please change the y-axis label of 4A to “outbreak size” so that it is consistent with the rest of the figures. “Total number of infections” can be mentioned in brackets as a definition.

Response: We have changed the Y-axis label of Figure 4A to “outbreak size”.

Reviewer 2:

Below are minor comments.

Abstract:

I much preferred the previous abstract. It was easier to read and had a nice flow to it. This abstract needs cleaning up. i.e.

“Most of the current diffusion models only consider direct interactions among individuals to build underlying infectious items transmission networks.”

1) I think it was fine to refer to these other models as “network models”.

2) “to build underlying infectious items transmission networks” is not clear. I think what you are trying to say (which I agree with) is that ‘Most network models only consider direct contacts capable of transmitting disease’

P 2. L41-52. There are a lot of abbreviations introduced, but not defined. I feel like much of this should be in the methods section.

P 3 Line 16, Fig1 : define ‘L’ in the text (as the blue circle) and in the figure. Or leave it out completely.

P 3. 57. Change ‘directed transmission’ to ‘direct transmission’

P 4. L7 “we integrate...” This sentence needs clarifying.

P4. 7-14. You use ‘proximity’ and ‘location L’ to represent the same think. Just use one.

P4. L42. change ‘an SPDT links’ to ‘SPDT links’. Make this change elsewhere as well (ie pg 5 L20, ‘an SPDT to a SPDT’).

P4. (2.8) Does ‘E’, overall exposure, include both direct and indirect links? As it is written, it looks like it only includes indirect links. Please clarify. Should E not include both E_d and E_i?

P4. L59. This is the first time you have mentioned virus. Until this point you have kept the infectious agent general. I would suggest continuing to do so, or state that you are modelling viral dynamics earlier.

P5 L24-26. “When the host user’s last...” This is an important sentence and needs cleaning up/ clarification.

P5 L 26-17. Perhaps change “of last update” to “for up to 200 min since the last update” (?)

P5. L 49-50. ‘diffusion behaviours of SPDT model comparing to SPST model’. change to: ‘of the SPDT model’ or ‘of SPDT models’... etc.

P6. Under Characterizing Metrics. The first sentence needs clarifying. I would not refer to I_n and I_p as parameters. Also, what is the symbol for individuals recovered? Is this I_r (which you use below?). Do the subscripts ‘n’ and ‘p’ implicitly represent time (simulation day)? Also can you elaborate on how you take the average of R_t to get R_e.

P7. L11. Is ‘r_t’ the same as ‘r’ defined above. Is the ‘t’ subscript indicative of decay time? I think you may have confused decay rate with the lifespan of a particle. I would think that as particles decay more quickly (ie the largest rate r_t=60), you would have fewer particles

lingering about, since they are decaying quickly. I am assuming that r_t is the lifespan, and r is the rate. Make this clean and relabel the figure.

P9 L10-12. "...reaches to the individuals,..."
change to "reaches the higher degree individuals"

P9 L55 subscript on R missing.

P11 consider using different symbol for recovery and infectivity. You use σ for both. (see top of P11 and (3.1). This becomes more confusing later on in your discussion on P14.

Our response to Reviewer 2

Abstract:

Comment-1: I much preferred the previous abstract. It was easier to read and had a nice flow to it. This abstract needs cleaning up. i.e.

"Most of the current diffusion models only consider direct interactions among individuals to build underlying infectious items transmission networks."

i) I think it was fine to refer to these other models as "network models".

ii) "to build underlying infectious items transmission networks" is not clear. I think what you are trying to say (which I agree with) is that 'Most network models only consider direct contacts capable of transmitting disease'

Response: We would like to thank Reviewer 2 for addressing these important issues. We have updated the abstract according to the suggestions. The revised Abstract of our manuscript as follow:

'Interaction patterns at the individual level influence the behaviour of diffusion over contact networks. Most of the current diffusion models only consider direct interactions, capable of transferring infectious items among individuals, to build transmission networks of diffusion. However, delayed indirect interactions, where a susceptible individual interacts with infectious items after the infected individual has left the interaction space, can also cause transmission events. We define a diffusion model called the same place different time transmission (SPDT) based diffusion that considers transmission links for these indirect interactions. Our SPDT model changes the network dynamics where the connectivity among individuals varies with the decay rates of link infectivity. We investigate SPDT diffusion behaviours by simulating airborne disease spreading on data-driven contact networks. The SPDT model significantly increases diffusion dynamics with a high rate of disease transmission. By making the underlying connectivity denser and stronger due to the inclusion of indirect transmissions, SPDT models are more realistic than SPST models for the study of various airborne diseases outbreaks. Importantly, we also find that the diffusion dynamics including indirect links are not reproducible by the current SPST models, even if both SPDT and SPST networks assume the same underlying connectivity. This is because the transmission dynamics of indirect links are different to those of direct links. The outcomes of this paper highlight the importance of the indirect links for predicting outbreaks.'

Comment-2 (P 2. L41-52): There are a lot of abbreviations introduced, but not defined. I feel like much of this should be in the methods section.

Response: Thanks to Reviewer 2 for suggesting to clarify the abbreviations of contact networks names used in our experiment. We have revised the text of Section 3.2 on p.5 to make clearer the definition of constructed networks. We have included some of them here.

'The above constructed networks show low link densities as users often appear in the system for an average of 3-4 days and then disappear for the remainder of the data collection period. This is characterised by the limitations of the collection system and user's behaviours when using the social networking App. Thus, these networks are called Sparse SPDT network and Sparse SPST network which capture partial snapshots of real-world social contact networks. In this paper, Sparse SPDT network is denoted as **SDT** network and Sparse SPST network as **SST** network.'

'We reconstruct a Dense SPDT network (**DDT** network) repeating links from available days of a user to the missing days for that user [2, 3]. In this process, all links of a day picked randomly from available days are copied to a random missing day. Thus, the DDT network has links for every day for each user. Then, the corresponding Dense SPST network (**DST** network) is built excluding indirect links from the DDT network.'

'We create two networks, **LDT** and **LST**, which maintain the same link densities and the same underlying social structure as that of the DDT network.'

Comment-3 (P 3 Line 16): Fig1 : define 'L' in the text (as the blue circle) and in the figure. Or leave it out completely.

Response: We have updated Figure 1 and the corresponding text as follows in Section 2 on p.3.

'In this particular scenario, an infected individual A (host individual), red circle, arrives at location L, blue dashed circle'

Comment-4 (P 3. 57): Change 'directed transmission' to 'direct transmission'

Response: We have replaced 'directed transmission' with 'direct transmission'

Comment-5 (P 4. L7): "we integrate..." This sentence needs clarifying.

Response: We have revised the text in the 3rd paragraph of Section 2 on p.3 of our manuscript.

'We combine a method with the SPDT model to assess the probability of contracting disease through an SPDT link (also called SPDT link infectivity) using generic assumptions'

Comment-6 (P4. 7-14): You use 'proximity' and 'location L' to represent the same think. Just use one.

Response: We have updated the relevant text in 4th paragraph of Section 2 on p.4 of our manuscript. The revised text is as follows:

'Suppose that an infected individual A appears at a location L at time t_s and deposits airborne infectious particles into the air of L with a rate g (particles/s)'

Comment-7 (P4. L42): change 'an SPDT links' to 'SPDT links'. Make this change elsewhere as well (ie pg 5 L20, 'an SPDT to a SPDT').

Response: We have revised the text accordingly

Comment-8 (P4, 2.8): Does 'E', overall exposure, include both direct and indirect links? As it is written, it looks like it only includes indirect links. Please clarify. Should E not include both E_d and E_i ?

Response: We would like to thank Reviewer 2 for seeking clarification for this import point. E is the summation of E_i while it is consisted of E_d and E_i . Therefore, E is consisted of both direct and indirect exposure. We have added some extra explanation to clarify it in 4th paragraph of Section 2 on p.4 of our manuscript. The revised text is as follows:

'where E_i^k is the received exposure for k^{th} SPDT link which have direct and/or indirect components'

Comment-9 (P4. L59): This is the first time you have mentioned virus. Until this point you have kept the infectious agent general. I would suggest continuing to do so, or state that you are modelling viral dynamics earlier.

Response: We have revised the text according to the suggestion. The last line of Section 2 on p.4 as follows:

'This value depends on both the disease type and the infectiousness of particles'

Comment-10 (P5 L24-26): "When the host user's last..." This is an important sentence and needs cleaning up/ clarification.

Response: We have rewritten the corresponding text in 1st paragraph of Section 3.2 on p.5 as follows:

'When the last update of the host user's is made from more than 20 metres distance of his first update of the current location or after 30 minutes of his immediate previous update, a

new link creation process is started at the new location.'

Comment-11 (P5 L 26-17): Perhaps change "of last update" to "for up to 200 min since the last update" (?)

Response: We have agreed with the Reviewer 2 and updated the relevant text in our manuscript as follows in 1st paragraph of Section 3.2 on p.5:

'However, the link creation is continued at the previous location for up to 200 minutes since the last update of the host user'

Comment-12 (P5. L 49-50): 'diffusion behaviours of SPDT model comparing to SPST model'. change to: 'of the SPDT model' or 'of SPDT models'... etc.

Response: We have made the suggested change in our manuscript.

Comment-13 P6: Under Characterizing Metrics. The first sentence needs clarifying. I would not refer to I_n and I_p as parameters. Also, what is the symbol for individuals recovered? Is this I_r (which you use below?). Do the subscripts 'n' and 'p' implicitly represent time (simulation day)? Also can you elaborate on how you take the average of R_t to get R_e .

Response: We would like to thank Reviewer 2 for noticing these important points regarding the characterising metrics in our manuscript. The subscripts 'n' and 'p' presents new infection and prevalence infection in the system. We have explained the calculation of R_e from R_t . We have revised the text reflecting this in Section 3.5 on p.6.

'We have collected the following values of disease incidents at each day of simulation to characterize diffusion dynamics in the networks: number of infections $I_n(t)$ caused at a simulation day t , number of infected individuals $I_r(t)$ recovered from infection and current number of infected individuals $I_p(t)$ (disease prevalence) in the system on a simulation day t .'

'Then, we find the average R_e of R_t summing R_t and dividing by the number of simulation days. The value of R_e represents the overall strength of a disease to diffuse on the contact networks.'

Comment-14 (P7. L11): Is ' r_t ' the same as ' r ' defined above. Is the ' t ' subscript indicative of decay time? I think you may have confused decay rate with the lifespan of a particle. I would think that as particles decay more quickly (ie the largest rate $r_t=60$), you would have fewer particles lingering about, since they are decaying quickly. I am assuming that r_t is the lifespan, and r is the rate. Make this clean and relabel the figure.

Response: We have revised the text to reflect the suggestion of Reviewer 2. In our experiment, r_t is the time corresponding to a r . In fact, through the value of r_t we represents r as r_t is more convenient to read while r is very small fractional value and reduces readability.

We have defined the r_t in the caption of Figure 2 on p.7 of our manuscript. So that reader can understand how r_t relates r .

'Particles concentrations for various particles decay rates r_t , where $r_t = 1/60r$ min and r is the proportion of particles is decayed per second, at a visited location when A) an infected individual is present at the location over 200 min and B) the infected individual has left the location after staying 200 min'

P9 L10-12. "...reaches to the individuals,..."
change to "reaches the higher degree individuals"

Response: We have replaced the text according to the suggestion of Reviewer 2.

P9 L55 subscript on R missing.

Response: We have fixed it.

P11 consider using different symbol for recovery and infectivity. You use σ for both. (see top of P11 and (3.1). This becomes more confusing later on in your discussion on P14.

Response: We have fixed it. We have used τ for recovery rate and σ for infectivity.